# On the Robustness of Neural Collapse and the Neural Collapse of Robustness

**Jingtong Su**                                                                *js12196@nyu.edu*
*Center for Data Science*
*New York University*

**Ya Shi Zhang**                                                              *ysz23@cam.ac.uk*
*Statistical Laboratory*
*University of Cambridge*

**Nikolaos Tsilivis**                                                          *nt2231@nyu.edu*
*Center for Data Science*
*New York University*

**Julia Kempe**                                                                *kempe@nyu.edu*
*Center for Data Science and Courant Institute of Mathematical Sciences*
*New York University*

**Reviewed on OpenReview:** *https://openreview.net/forum?id=OyXS4ZIqd3*

## Abstract

Neural Collapse refers to the curious phenomenon in the end of training of a neural network, where feature vectors and classification weights converge to a very simple geometrical arrangement (a simplex). While it has been observed empirically in various cases and has been theoretically motivated, its connection with crucial properties of neural networks, like their generalization and robustness, remains unclear. In this work, we study the stability properties of these simplices. We find that the simplex structure disappears under small adversarial attacks, and that perturbed examples "leap" between simplex vertices. We further analyze the geometry of networks that are optimized to be robust against adversarial perturbations of the input, and find that Neural Collapse is a pervasive phenomenon in these cases as well, with clean and perturbed representations forming aligned simplices, and giving rise to a robust simple nearest-neighbor classifier. By studying the propagation of the amount of collapse inside the network, we identify novel properties of both robust and non-robust machine learning models, and show that earlier, unlike later layers maintain reliable simplices on perturbed data. Our code is available at https://github.com/JingtongSu/robust_neural_collapse.

## 1 Introduction

Deep Neural Networks are nowadays the de facto choice for a vast majority of Machine Learning applications. Their success is often attributed to their ability to jointly learn rich feature functions from the data and to predict accurately based on these features. While the exact reasons for their generalization abilities still remain elusive despite a profusion of active research, they can, at least partially, be explained by implicit properties of the training algorithm, i.e some variant of gradient descent, that specifically biases the solutions towards having certain geometric properties (such as the maximum possible separation of the training points)(Neyshabur et al., 2015; Soudry et al., 2018).

Reinforcing arguments about the simplicity of the networks found by stochastic gradient descent in classification settings, Papyan et al. (2020) made the surprising empirical observation that both the feature representations

in the penultimate layer (grouped by their corresponding class) and the weights of the final layer form a *simplex equiangular tight frame* (ETF) with $C$ vertices, where $C$ is the number of classes. Curiously, such a geometric arrangement becomes more pronounced well-beyond the point of (effectively) zero loss on the training data, motivating the common tendency of practitioners to optimize a network for as long as the computational budget allows. The collection of these empirical phenomena was termed *Neural Collapse*.

While the results of Papyan et al. (2020) fueled much research in the field, many questions remain regarding the connection of Neural Collapse with properties like generalization and robustness of Neural Networks. In particular with regards to *adversarial robustness*, the ability of a model to withstand adversarial modifications of the input without effective drops in performance, it has been originally claimed that the instantiation of Neural Collapse has positive effect on defending against adversarial attacks (Papyan et al., 2020; Han et al., 2022). However, this seems to at least superficially contradict the fact that neural networks are not a priori adversarially robust (Szegedy et al., 2014; Carlini and Wagner, 2017).

**Our contributions.** Through an extensive empirical investigation with computer vision datasets, we study the robustness of the formed simplices for several converged neural networks. Specifically, we find that gradient-based adversarial attacks with standard hyperparameters alter the feature representations, resulting in neither variability collapse nor simplex formation. To decouple our analysis from label dependencies that accompany untargeted attacks, we further perform *targeted attacks* that maintain class balance, and show that perturbed points end up almost exactly in the target class-mean vertex of the original simplex, meaning that even this optimal simplex arrangement is quite fragile.

Moving forward, we pose the question of whether Neural Collapse can appear in other settings in Deep Learning, in particular in robust training settings. Interestingly, we find that Neural Collapse also happens during adversarial training (Goodfellow et al., 2015; Madry et al., 2018), a worst-case version of empirical risk minimization (Madry et al., 2018), and, in fact, simplices form both for the "clean" original samples and the adversarially perturbed training data. Curiously, the amount of collapse and simplex formation is much less prevalent when alternative robust training methods (Zhang et al., 2019) are deployed (with the same ability to fit the training data).

Finally, we study the geometry of the inner layers of the network through the lens of Neural Collapse, and analyze the propagation of representations of both natural and adversarial data. We observe two novel phenomena inside standardly trained (non-robust) networks: (a) feature representations of adversarial examples in the earlier layers show some form of collapse which disappears later in the network, and (b) the nearest-neighbors classifiers formed by the class-centers of feature representations that correspond to early layers have significant accuracy on adversarial examples of the network.

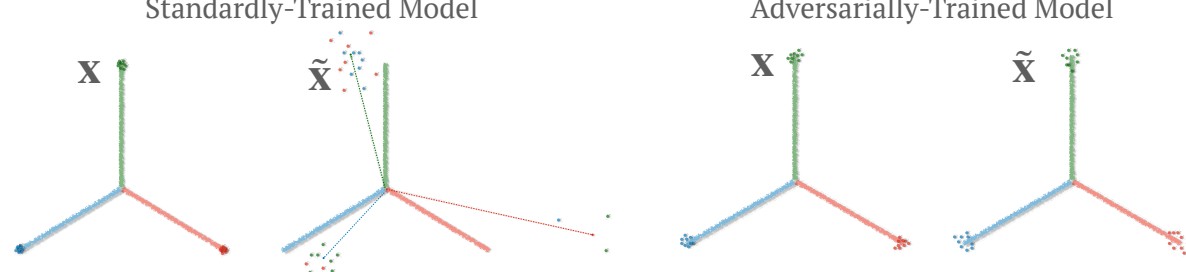

Figure 1: Visualisation of our findings. Sticks represent clean class-means. Small dots correspond to the representation of an individual datum. The color represents the ground-truth label, and the dotted lines represent the predicted class-means. **Left to Right:** clean representations with standardly-trained (ST) networks; perturbed representations with standardly-trained networks; clean representations with adversarially-trained (AT) networks; perturbed representations with adversarially-trained networks. With ST nets, the adversarial perturbations push the representation to "leap" towards another cluster with slight angular deviation. AT makes the simplex resilient to such adversarial attacks, with higher and intra-class variance.

To summarize, our contributions and findings are the following, partly illustrated in Figure 1:

- **Is NC robust?** We initiate the study of the neural collapse phenomenon in the context of adversarial robustness, both for standarly trained networks under adversarial attacks and for adversarially trained robust networks to investigate the stability and prevalence of the NC phenomenon. Our work exposes considerable additional fundamental, and we think, surprising, geometrical structure:

- **No!** For standarly trained networks we find that small, imperceptible adversarial perturbations of the training data remove any simplicial structure at the representation layer: neither variance collapse nor simplex representations appear under standard metrics. Further analysis through class-targeted attacks that preserve class-balance shows a "cluster-leaping" phenomenon: representations of adversarially perturbed data jump to the (angular) vicinity of the original class means.

- **Yes for AT networks! Two identical simplices emerge.** Adversarially trained, robust, networks exhibit a simplex structure both on original clean and adversarially perturbed data, albeit of higher variance. These two simplices turn out to be the same. We find that the simple nearest-neighbor classifiers extracted from such models also exhibit robustness.

- **Early layers are more robust.** Analyzing NC metrics in the representations of the inner layers, we observe that initial layers exhibit a higher degree of collapse on adversarial data. The resulting simplices, when used for Nearest Neighbor clustering, give surprisingly robust classifiers. This phenomenon disappears in later layers.

The outline of this paper is as follows: In Section 2 we summarize previous work, in Section 3 we describe the measures and methods we use, Section 4 gives our experimental results and the insights we derive from them, and Section 5 concludes.

## 2 Related work

**Neural Collapse & Geometric properties of Optimization in Deep Learning.** The term Neural Collapse was coined by Papyan et al. (2020) to describe phenomena about the feature representations of the last layer and the classification weights of a deep neural network at *convergence*. It collectively refers to the onset of variability collapse of within-class representations (NC1), the formation of two simplices (NC2) - one from the class-mean representations and another from the classification weights - that are actually dual (NC3), and, finally, the underlying simplicity of the prediction rule of the network, which becomes nothing but a simple nearest-neighbor classifier (NC4) (see Section 3 for formal definitions). Papyan et al. (2020), using ideas from Information Theory, showed that the formation of a simplex is optimal in the presence of vanishing within-class variability. Mixon et al. (2022) introduced the *unconstrained features* model (independently proposed by Fang et al. (2021) as the Layer-Peeled model), a model where the feature representations are considered as free optimization variables, and showed that a global optimizer of this problem (for the MSE loss) exhibits Neural Collapse. Many derivative works have proven Neural Collapse modifying this model, by either considering other loss functions or trying to incorporate more deep learning elements into it (Fang et al., 2021; Zhu et al., 2021; Ji et al., 2022; E and Wojtowytsch, 2022; Zhou et al., 2022; Tirer and Bruna, 2022; Han et al., 2022). The notion of maximum separability dates back to Support Vector Machines (Cortes and Vapnik, 1995), while the bias of gradient-based optimization algorithms towards such solutions has been used to explain the success of boosting methods (Schapire et al., 1997), and, more recently, to motivate the generalization properties of neural networks (Neyshabur et al., 2015; Soudry et al., 2018; Lyu and Li, 2020). The connection between Neural Collapse and generalization of neural networks (on in-distribution and transfer-learning tasks) has been explored in Galanti et al. (2021); Hui et al. (2022). Finally, the propagation of Neural Collapse inside the network has been studied by Ben-Shaul and Dekel (2022); He and Su (2022); Hui et al. (2022); Li et al. (2022); Tirer et al. (2022); Rangamani et al. (2023).

**Adversarial Examples & Robustness.** Neural Networks are famously susceptible to adversarial perturbations of their inputs, even of very small magnitude (Szegedy et al., 2014). Most of the attacks that drive the performance of networks to zero are gradient-based (Goodfellow et al., 2015; Carlini and Wagner, 2017). These perturbations are surprisingly consistent between different architectures and hyperparameters, they are in many cases transferable between models (Papernot et al., 2017), and they can also be made universal

(one perturbation for all inputs) (Moosavi-Dezfooli et al., 2017). For training robust models, one can resort to algorithms from robust optimization (Xu et al., 2009; Goodfellow et al., 2015; Madry et al., 2018). In particular, the most effective algorithm used in deep learning is called *Adversarial Training* (Madry et al., 2018). During adversarial training one alternates steps of generating adversarial examples and training on this data instead of the original one. Several variations of this approach have been proposed in the literature (e.g. Zhang et al. (2019); Shafahi et al. (2019); Wong et al. (2020)), modifying either the attack used for data generation or the loss used to measure mistakes. However, models produced by this algorithm, despite being relatively robust, still fall behind in terms of absolute performance (Croce et al., 2021), while there are still many unresolved conceptual questions about adversarial training (Rice et al., 2020a). In terms of the geometrical properties of the solutions, Li et al.; Lv and Zhu (2022) showed that in some cases (either in the presence of separable data or/and homogeneous networks) adversarial training converges to a solution that maximally separates the *adversarial* points.

## 3 Background & Methodology

In this section, we proceed with formal definitions of Neural Collapse (NC), adversarial attacks, and Adversarial Training (AT), together with the variants we study in this paper.

### 3.1 Notation

Let $\mathcal{X}$ be an input space, and $\mathcal{Y}$ be an output space, with $|\mathcal{Y}| = C$. Denote by $\mathcal{S}$ a given class-balanced dataset that consists of $C$ classes and $n$ data points per class. Let $f : \mathcal{X} \to \mathcal{Y}$ be a neural network, with its final linear layer denoted as $\mathbf{W}$. For each class $c$, the corresponding classifier — i.e. the $c$-th row of $W$ — is denoted as $\mathbf{w}_c$, and the bias is called $b_c$. Denote the representation of the $i$-th sample within class $c$ as $\mathbf{h}_{i,c} \in \mathbb{R}^p$, and the union of such representations $H(\mathcal{S})$. We define the global-mean vector $\boldsymbol{\mu}_G \in \mathbb{R}^p$, and class-mean vector $\boldsymbol{\mu}_c \in \mathbb{R}^p$ associated with $\mathcal{S}$ as $\boldsymbol{\mu}_G \triangleq \frac{1}{nC} \sum_{i,c} \mathbf{h}_{i,c}$ and $\boldsymbol{\mu}_c \triangleq \frac{1}{n} \sum_i \mathbf{h}_{i,c}, c = 1, \ldots, C$. For brevity, we refer in the text to the globally-centered class-means, $\{\boldsymbol{\mu}_c - \boldsymbol{\mu}_G\}_{c=1}^C$, as just *class-means*, since these vectors are constituents of the simplex. We denote $\tilde{\boldsymbol{\mu}}_c = (\boldsymbol{\mu}_c - \boldsymbol{\mu}_G)/\|\boldsymbol{\mu}_c - \boldsymbol{\mu}_G\|_2$ the normalized class-means. Unless otherwise specified, the term "representation" refers to the penultimate layer of the network. Additionally, we will refer to class-means of representations of adversarial examples (see Section 3.3) as 'perturbed class-means.' This is because adversarially robust networks can still achieve 100% training accuracy on 'adversarial' perturbations of the original dataset $S$.

### 3.2 Neural Collapse Concepts

Papyan et al. (2020) demonstrate the prevalence of NC on networks optimized by SGD in the *Terminal Phase of Training* (TPT) – the phase beyond the point of zero training error and towards zero training loss – by tracing the following quantities. Throughout our paper, we closely follow Papyan et al. (2020) and Han et al. (2022) on formalization of NC1-NC4. Please refer to the Appendix A for exact definitions.

**(NC1) Variability collapse:** For all classes $c$, the within-class variation collapses to zero:
$$\mathbf{h}_{i,c} \to \boldsymbol{\mu}_c \quad \forall\, i \in [n], c \in [C].$$

**(NC2, Equiangular) Class-Means converge to Equal, Maximally-Separated Pair-wise Angles:** For every pair of distinct labels $c \neq c'$,
$$\langle \tilde{\boldsymbol{\mu}}_c, \tilde{\boldsymbol{\mu}}_{c'} \rangle \to -\frac{1}{C-1}.$$

**(NC2, Equinorm) Class-Means Converge to Equal Length:**
$$|\|\boldsymbol{\mu}_c - \boldsymbol{\mu}_G\|_2 - \|\boldsymbol{\mu}_{c'} - \boldsymbol{\mu}_G\|_2| \to 0 \quad \forall\, c, c'.$$

**(NC3) Convergence to self-duality:** The linear classifier $\mathbf{W}$ and the class-means converge to each other[1]:
$$\frac{\mathbf{w}_c}{\|\mathbf{w}_c\|_2} - \frac{\boldsymbol{\mu}_c - \boldsymbol{\mu}_G}{\|\boldsymbol{\mu}_c - \boldsymbol{\mu}_G\|_2} \to 0 \quad \forall\, c.$$

---

[1]Using the definition of NC3 from Han et al. (2022).

**(NC4): Simplification to Nearest Class Center (NCC) classifier:** The prediction of the network is equivalent to that of the NCC classifier formed by the non-centered class-means on $\mathcal{S}$:

$$\arg\max_{c'} \langle \mathbf{w}_{c'}, \mathbf{h} \rangle + b_{c'} \to \arg\min_{c'} \|\mathbf{h} - \boldsymbol{\mu}_{c'}\|_2 \quad \forall \, \mathbf{h} \in H(\mathcal{S}).$$

In this work, we compare representations of original and perturbed data, which imposes ambiguity on which class-mean vectors $\boldsymbol{\mu}_c, \boldsymbol{\mu}_G$ to use (from $\mathcal{S}$ or $\mathcal{S}'$). In the spirit of the original definitions, for NC1-4 we will use the class means induced by the dataset $\mathcal{S}'$, even if different from the training set. NC4 studies the predictive power of the NCC classifier on $\mathcal{S}'$ by comparing it to the network classification output, which at TPT is equivalent to the ground truth label. For study of reference data $\mathcal{S}'$ outside the TPT, we introduce two quantities, which we use in Section 4 to study Neural Collapse in intermediate layers:

**NCC-Network Matching Rate:** It measures the rate at which the NCC classifier defined in NC4 trained on $\mathcal{S}$ coincides with the output of the network on dataset $\mathcal{S}'$. Note that we use $\boldsymbol{\mu}_c$ calculated by $\mathcal{S}$.

$$\arg\max_{c'} \langle \mathbf{w}_{c'}, \mathbf{h} \rangle + b_{c'} \overset{?}{=} \arg\min_{c'} \|\mathbf{h} - \boldsymbol{\mu}_{c'}\|_2, h \in H(\mathcal{S}').$$

**NCC Accuracy:** It measures the accuracy on dataset $\mathcal{S}'$ of the NCC classifier defined in NC4 trained on $\mathcal{S}$. Note that we use $\boldsymbol{\mu}_c$ calculated by $\mathcal{S}$. $c_h$ denotes the ground-truth label of the input.

$$c_h \overset{?}{=} \arg\min_{c'} \|\mathbf{h} - \boldsymbol{\mu}_{c'}\|_2, h \in H(\mathcal{S}').$$

Note that when $\mathcal{S} = \mathcal{S}'$, both NCC-Network Matching Rate and NCC Accuracy stem from (NC4). We also introduce the following measures to quantify the proximity of two simplices over $C$-classes:

**Simplex Similarity:** We define the similarity measure between two $C$-class simplices with normalized class means $\tilde{\boldsymbol{\mu}}_c, \tilde{\boldsymbol{\mu}}_c'$ as

$$\mathrm{AVG}_c \arccos \langle \tilde{\boldsymbol{\mu}}_c, \tilde{\boldsymbol{\mu}}_c' \rangle.$$

**Non-centered Angular Distance:** Similarly, given two simplices, without taking the global mean $\boldsymbol{\mu}_G$ and $\boldsymbol{\mu}_G'$ into account, we can calculate the angular distance with non-centered class-means directly:

$$\mathrm{AVG}_c \arccos \langle \frac{\boldsymbol{\mu}_c}{\|\boldsymbol{\mu}_c\|_2}, \frac{\boldsymbol{\mu}_c'}{\|\boldsymbol{\mu}_c'\|_2} \rangle.$$

Note that the similarity and angular distance between a simplex and itself is zero.

## 3.3 Gradient-Based Adversarial Attack, Adversarial Training (AT), and TRADES

Given a deep neural network $f$ with parameters $\theta$, a clean example $(\mathbf{x}, y)$ and cross-entropy loss $\mathcal{L}(\cdot, \cdot)$, the *untargeted* adversarial perturbation is crafted by running multiple steps of projected gradient descent (PGD) to maximize the CE loss (Kurakin et al., 2017; Madry et al., 2018) (in what follows, we focus on $\ell_\infty$ adversary with $\ell_2$ deferred to the appendix):

$$\mathbf{x}^{k+1} = \Pi_{\mathcal{B}_{\mathbf{x}^0}^\epsilon} \left( \mathbf{x}^k + \alpha \cdot \mathrm{sign}(\nabla_{\mathbf{x}^k} \mathcal{L}(f(\mathbf{x}^k), y)) \right), \tag{1}$$

where $\mathbf{x}^0 = \mathbf{x}$ is the original example, $\alpha$ is the step size, $\tilde{\mathbf{x}} = \mathbf{x}^N$ is the final adversarial example, and $\Pi$ is the projection on the valid $\epsilon$-constraint set, $\mathcal{B}_{\mathbf{x}}^\epsilon$, of the data. $\mathcal{B}_{\mathbf{x}}^\epsilon$ is usually taken as either an $\ell_\infty$ or $\ell_2$ ball centered in $\mathbf{x}^0$. Further, to control the predicted label of $\tilde{\mathbf{x}}$, a variant called *targeted attack* minimizes the CE loss w.r.t. a target label $y_t \neq y$:

$$\mathbf{x}^{k+1} = \Pi_{\mathcal{B}_{\mathbf{x}^0}^\epsilon} \left( \mathbf{x}^k - \alpha \cdot \mathrm{sign}(\nabla_{\mathbf{x}^k} \mathcal{L}(f(\mathbf{x}^k), y_t)) \right). \tag{2}$$

With a standardly-trained network, both these methods can effectively reduce the accuracy to 0%. To combat this phenomenon, robust optimization algorithms have been proposed. The most representative methodology, *adversarial training* (Madry et al., 2018), generates $\tilde{\mathbf{x}}$ *on-the-fly* with Equation (1) for each epoch from $\mathbf{x}$, and takes the model-gradient update on $\tilde{\mathbf{x}}$ only.

An alternative robust training variant, TRADES (Zhang et al., 2019), is of particular interest as it aims to address both robustness and clean accuracy. Thus the gradient steps of TRADES directly involve both $\mathbf{x}$ and $\bar{\mathbf{x}}$, where $\bar{\mathbf{x}}$ is also obtained by PGD, but under the KL-divergence loss:

$$\mathbf{x}^{k+1} = \Pi_{\mathcal{B}_{\mathbf{x}^0}^\epsilon} \left( \mathbf{x}^k + \alpha \cdot \mathrm{sign}(\nabla_{\mathbf{x}^k} \mathcal{L}_{KL}(f(\mathbf{x}), f(\mathbf{x}^k))) \right). \tag{3}$$

The total TRADES loss is a summation of the CE loss on the clean data and a KL-divergence (KLD) loss between the predicted probability of $\mathbf{x}$ and $\bar{\mathbf{x}}$ with a regularization constant $\beta$:

$$\mathcal{L}_{CE}(f(\mathbf{x}), y) + \beta \cdot \mathcal{L}_{KL}(f(\mathbf{x}), f(\bar{\mathbf{x}})). \tag{4}$$

## 4 Experiments

In this section, we present our main experimental results measuring neural collapse in standardly (ST) and adversarially (AT) trained models. When collecting feature representations for adversarially perturbed data we always compute the epoch-relevant perturbations: for ST models throughout training we compute NC metrics for data perturbed relative to the model at the current training epoch. For AT models at each epoch we use the current (adversarially perturbed) training data.

**Datasets** We consider image classification tasks on CIFAR-10, CIFAR-100 in our main text. Both datasets are balanced (in terms of images per class), so we comply with the original experimental setup of Papyan et al. (2020). We preprocess the images by subtracting their global (train) mean and dividing by the standard deviation. For the completeness of our experiments, we also consider a 10-class subset of ImageNet: ImageNette[2]. We pick the 160px variant. The results are presented in Appendix F.

**Models** We train two large convolutional networks, a standard VGG and a Pre-Activation ResNet18, from a random initialization. Both models have sufficient capacity to fit the entire training data. We launch 3 independent runs and report the mean and standard deviation throughout our paper.

**Algorithms** We train the networks using stochastic gradient descent, either optimizing the cross entropy loss (standard training - ST) or the worst case loss, bounded by either an $\ell_2$ or $\ell_\infty$ perturbation (adversarial training - AT). For CIFAR-10/100 datasets, we adopt community-wide standard values for the perturbations following Rice et al. (2020a): for the $\ell_\infty$ adversary, we use radius $\epsilon = 8/255$ and step size $\alpha = 2/255$ in Equation 1. For the $\ell_2$ adversary, we use radius $\epsilon = 128/255$ and step size $\alpha = 15/255$. We perform 10 PGD iterations. All networks are being trained for 400 epochs in order to reach the terminal phase of training (post zero-error), with batch size 128 and initial learning rate 1e-1. We drop the learning rate by a factor of 0.1 at the 100th and again at the 150th epoch. We also consider the TRADES algorithm (Zhang et al., 2019) with Equation (4), setting $\beta = 6$ following Zhang et al. (2019). For ImageNette, we pick $\ell_2$ radius $\epsilon = 1536/255$ and step size $\alpha = 360/255$, and the same $\ell_\infty$ hyperparameters as for the CIFAR family. We perform 5 PGD iterations due to the larger image size. For full experimental details, please refer to Appendix B.

**Gaussian perturbation benchmark** To disentangle the effect of the size of the perturbations and their adversarial nature, we benchmark our NC analysis with "Gaussian" perturbations randomly drawn from $\mathcal{N}(0, (8/255)^2)$, i.e. of the same variance as the adversarial perturbation.

We present results for standardly trained models and $\ell_\infty$-adversarially trained ones on CIFAR-10 in the main text and defer the rest of the results to the Appendix C and D, with similar conclusions. In Appendix E, we also study NC phenomena under smaller adversarial $\ell_\infty$ perturbations and smaller radii in adversarial training (2/255, 4/255 and 6/255); the results interpolate as one would expect.

Previous research has observed that the neural collapse phenomenon under standard training does *not* hold on the *test set* (e.g., Hui et al. (2022).) We investigated the evolution of accuracy, loss, and neural collapse metrics for both clean and adversarially perturbed test set data. We not only reproduce the non-collapsed dynamic with ST, but also observe an even more severe non-collapse phenomenon on the test set under robust optimization. Please refer to Appendix G for the results.

### 4.1 Standardly trained neural nets

**Instability of Simplices** The first and third column of Figure 2 show the evolution of the NC quantities as described in Section 3 for standardly trained models. We use both adversarially perturbed and Gaussian reference data to study the stability of the original simplices. As expected, NC metrics converge on the clean training data. For Gaussian reference data, Neural Collapse is only slightly attenuated and is hardly

---

[2]https://github.com/fastai/imagenette

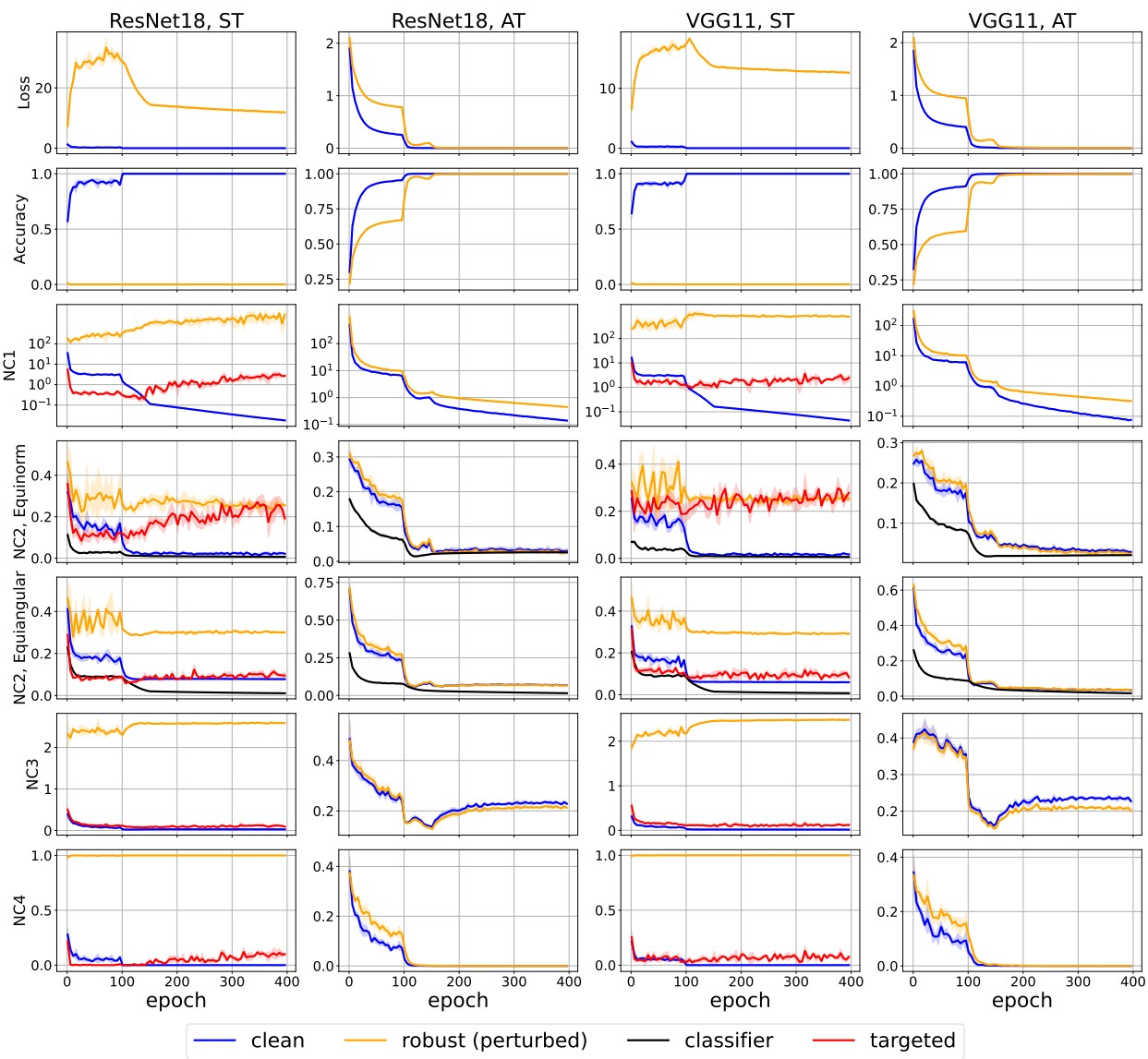

Figure 2: Accuracy, Loss, and NC evolution for standardly (ST) and adversarially (AT) trained VGG and ResNet. For AT models, clean and Guassian curves coincide. Setting: CIFAR-10, $\ell_\infty$ adversary.

distinguishable from the clean curve, thus we choose to omit it. Strikingly, NC disappears for adversarially perturbed data. These findings suggest that the simplex formed by clean training data is robust against random perturbations, but fragile to adversarial attacks. The results certainly corroborate the conclusion that the representation class-means of perturbed points with ground-truth label $c$ do not form any geometrically-meaningful structure at all.

**Re-emergence of Simplices: Cluster Leaping** To gain more insight into the geometric structure of the representations of adversarial examples, we propose modifications to our metrics to understand whether any simplex-like structure on adversarial data disappears or if we could retrieve simplicial remnants. We thus ask whether labeling adversarial data with *predicted* labels instead of ground truth class labels might lead to re-emergence of a geometrical structure. Note that classes formed by predicted labels are not any more balanced in general (see left column of Figure 3) and in fact might have vanishing classes, especially for larger datasets like CIFAR-100. Since the elegant simplex-structure of NC is predicated on balanced classes, NC metrics as defined will be hampered by this imbalance. To gain some coarse insight into the

geometry - leveraging the existence of a simplex ETF with ST when centering with the global-mean vector $\boldsymbol{\mu}_G$ - in Figure 3 we study perturbed representations *relative to* the original training-data (clean) simplex, by measuring how predicted class-means deviate from the corresponding clean class-means when centering them with the same clean global-mean vector $\boldsymbol{\mu}_G$. We outline two key insights: First, the predicted class-means have varying norms. Second, the angular distance between each pair of clean and predicted class-means is in general small, as conceptually illustrated in Figure 1. There, all correctly predicted clean data are wrongly classified after the adversarial perturbation is applied. Further, the predicted class-means have varying norms whilst keeping a small angular distance to the clean class-means, which are represented by the dotted lines and sticks, respectively. We conclude that adversarial attacks manipulate the feature space in an intriguing way by pushing the representation from the ground-truth class-mean to the (wrongly) predicted class-mean with very small angular deviation ("cluster leaping"). This observation is non-trivial, since adversarial attacks are performed by maximizing the CE loss w.r.t. the true label. Intuitively, a successful attack should push the representation to be *not* aligned with the last layer's true label weight vector, but whether it would align or not with a wrong class's weight vector is not a prior clear. Our experiments answer this question in the affirmative.

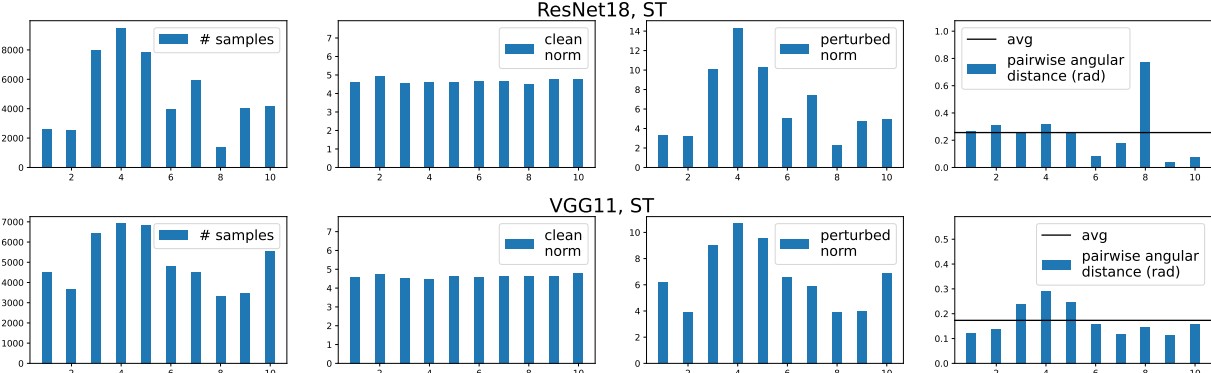

Figure 3: Illustration of untargeted adversarial attacks on standardly trained, converged, models that correspond to one random seed. (CIFAR-10, $\ell_\infty$). *Left:* Number of examples with a certain predicted label. *Inner Left:* The norms of clean class-means. *Inner Right:* The norms of predicted class-means with perturbed data. *Right:* Angular distance between clean and predicted class-mean with perturbed data. *Upper:* ResNet18; *Lower:* VGG11. For 10 classes, the between-class angular distance is $\arccos\left(-\frac{1}{9}\right) = 1.68$ rad $= 96.38$ degrees, while 0.2 rad is only 11.4 degrees.

**Targeted perturbations: merging simplices:** To further understand the geometry of representations of perturbed data and circumvent the class-imbalance issue, we perform *targeted attacks* (Equation (2)) in a circular way (samples of class $i$ are perturbed to resemble class $(i+1) \mod C$). Note that targeted attacks still result in 100% attack success rate.

NC metrics for targeted perturbations are shown in red in Figure 2. As already hinted at by results from the untargeted attack, the NC2 Equiangular term collapses, but NC2 Equinorm does not. This illustrates that the targeted class-means also form a maximally-separated geometrical structure but with varying norms on CIFAR-10. Also, the non-converging NC1 indicates that within-class predicted representations have oscillating positions. Further, from the vanishing NC3, we can infer that for each class $c$, these non-centered predicted class-mean vectors are very close in the angular space to the non-centered clean class-means $\boldsymbol{\mu}_c$, as they both approach $\mathbf{W}_c/||\mathbf{W}_c||$, implying that clean and predicted simplices are close. To provide additional verification beyond NC3, in Figure 4 (left subplots) we measure Simplex Similarity and non-centered Angular Distance of the simplices formed by targeted adversarial examples and by clean examples as described in Section 3. These results give us a full glimpse of how standardly trained networks are non-robust and fail under adversarial attacks: adversarial perturbations break the simplex ETF by "leaping" the representation from one class-mean to another, forming a norm-imbalanced less concentrated structure around the original simplex.

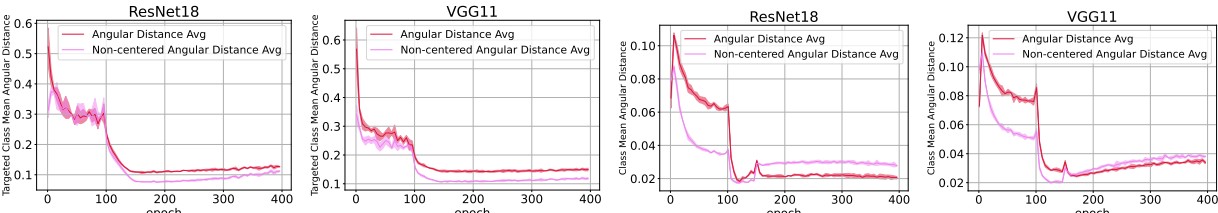

Figure 4: Angular distance. *Left and Inner Left:* Average between targeted attack class-means and clean class-means on **ST** network. *Inner Right and Right:* Average between perturbed class-means and clean class-means on **AT** network. Setting: CIFAR-10, $\ell_\infty$ adversary.

## 4.2 Neural Collapse during Adversarial Training

We train neural nets adversarially to full convergence with perfect clean and robust training accuracy and measure NC metrics for clean (original) and perturbed (epoch-wise) training data in Figure 2 (columns 2 and 4). Interestingly, we find that Neural Collapse *qualitatively* occurs in this setting as well, both for clean and perturbed data, and two simplices emerge. In particular, we find that for both ResNet18 and VGG, as robust training loss is driven to zero the NC metrics decrease on both the original (clean) train images and the perturbed points that we use on each epoch to update the parameters. Notice, however, that the extent of variability collapse (NC1) on the perturbed points is smaller than on the "clean" data or the Gaussian noise benchmark, indicating that clean examples are more concentrated around the vertices. To understand the relative positioning of the two simplices, we investigate the Simplex Similarity and Angular Distance between non-centered class-means in Figure 4 (right). The vanishing distance confirms these two simplices are exactly the same. We notice that the angular distance with AT grows initially and then gradually drops. We explain this phenomenon as follows. At the initial phase, the network's accuracy is only slightly higher than its robustness (see e.g., Figure 2) with both terms relatively low. Thus, untargeted attacks are not capable of modifying the features away from clean class-means yet. Gradually, as clean accuracy rises and clean clusters start to emerge, adversarial attacks become more and more successful, and thus we observe an increase in the angular distance between the classes.

These results suggest that Adversarial Training nudges the network to learn simple representational structures (namely, a simplex ETF) not only on clean examples but also on perturbed examples to achieve robustness against adversarial perturbations. Equivalently, the simplices induced by robust networks are *not fragile* anymore, but *resilient*. Note also that NC4 results imply that there is a simple nearest-neighbor classifier that is robust against adversarial perturbations generated from the network.

## 4.3 No Neural Collapse under TRADES Objective

One could conjecture that the formation of two very close simplices in robust models are necessary for robustness. Curiously, this is not the case for all training algorithms that produce robust models. In particular, Figure 5 demonstrates that a state-of-the-art algorithm that aims to balance clean and robust accuracy, TRADES (Zhang et al., 2019), shows fundamentally different behavior. Even though both terms of the loss (see Equation 4) are driven to zero, we do not observe Neural Collapse either qualitatively or quantitatively; the amount of collapse on both clean and perturbed data is roughly one order of magnitude larger than for adversarial training, and the feature representations do not approach the ETF formation, even well past the onset of the terminal phase. We view this as evidence that the prevalence of Neural Collapse is not necessary for robust classification.

## 4.4 Neural Collapse in Early Layers and Early-Stage Robustness

While originally variability collapse and simplex formation were observed for the last layer representations, follow-up studies extended the analysis to the intermediate layers of the neural network. In particular, He and Su (2022) found that the amount of variability collapse measured at different layers (at convergence) decreases

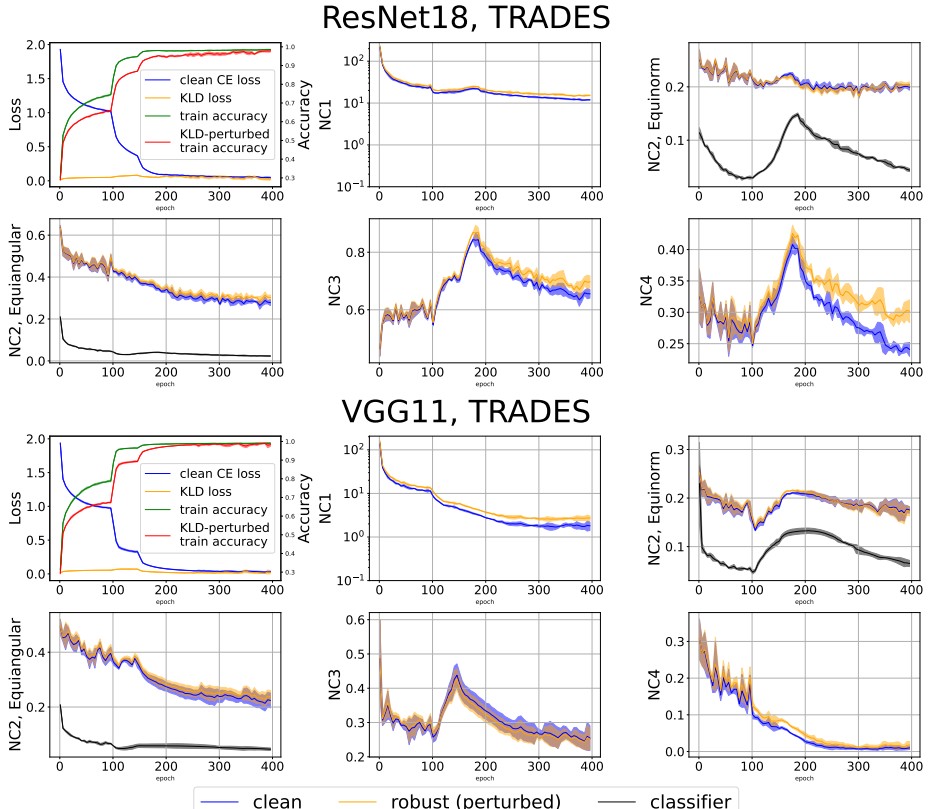

Figure 5: Accuracy, Loss and NC evolution with TRADES trained networks. *Upper:* ResNet18; *Lower:* VGG11. No simplices are formed with TRADES training. Setting: CIFAR-10, $\ell_\infty$ adversary. Note that we plot the KLD-loss here to showcase optimization convergence, to avoid the effect of the regularization constant $\beta$.

smoothly as a function of the index of the layer. Further, Hui et al. (2022) coined the term Cascading Neural Collapse to describe the phenomenon of cascading variability collapse; starting from the end of the network, the collapse of one layer seemed to be signaling the collapse of the previous layers (albeit to a lesser extent). Here, we replicate this study of the intermediate layer computations, while also studying the representations of the perturbed points (both in standard and adversarial training). In particular, we collect the input of either convolutional or linear layers of the network *at convergence*, order them by depth index, and compute the NC quantities of Section 3. The results are presented in Figure 6.

Both for ST and AT models, we reproduce the power law behavior observed in He and Su (2022) for clean data; the feature variability collapses progressively, and, interestingly, undergoes a slower decrease in the case of adversarial training. The adversarial data representations for ST models, however, while failing to collapse at the final layer (as already established in Figure 2), exhibit the same extent of Neural Collapse as those of the original data for the earlier layers. This hints that from the viewpoint of the earlier layers, clean and adversarial data are indistinguishable. And, this, is indeed the case! Looking at the first and third column of Figure 7, we observe that the simple classifier formed by the centers of the early layers is quite robust ($\sim 40\%$) to these adversarial examples (both train and test). Curiously, this robustness is higher than the one of the simple classifiers defined by layers of an adversarially trained model (although the two numbers are not directly comparable). This is, undeniably, a peculiar phenomenon of standardly trained models that is worth more exploration; could it be that the lesser variability exhibited in the earlier layers is actually beneficial for robustness or is it just the stability of the feature space that makes prediction more robust?

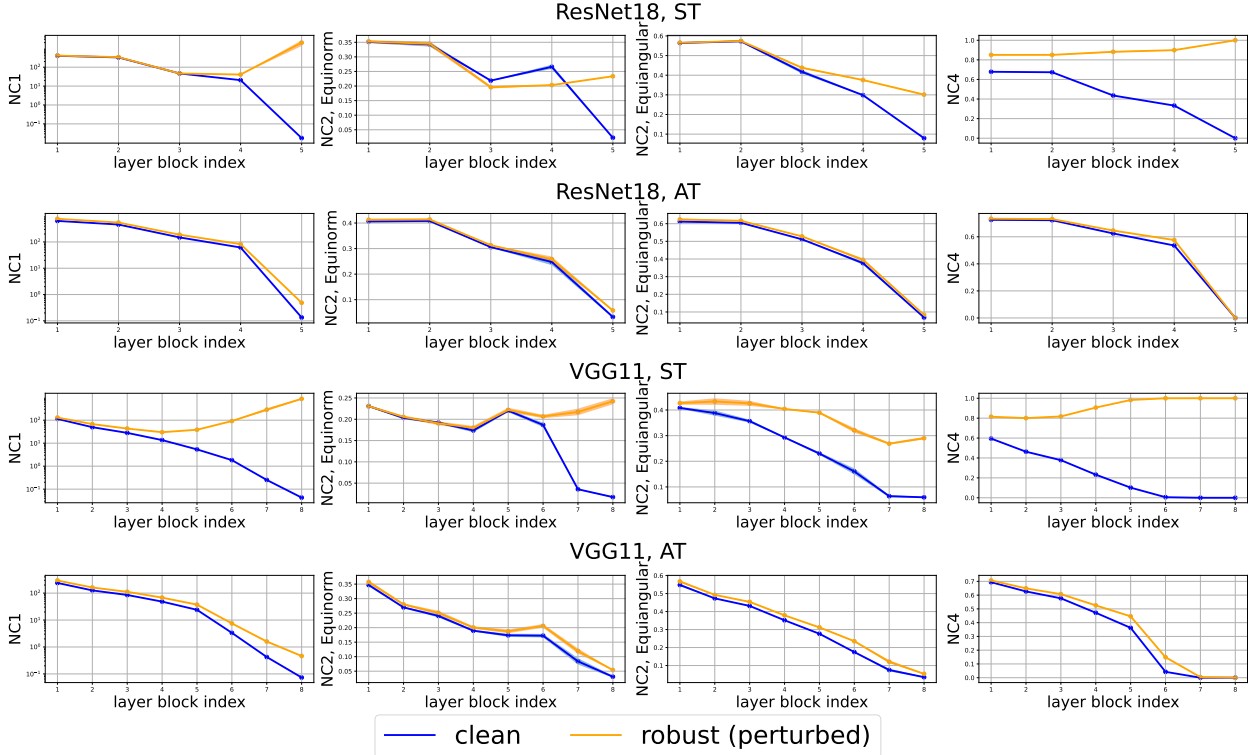

Figure 6: Layerwise evolution of NC1, NC2 and NC4 for ST and AT networks. NC metrics for perturbed data tend to undergo some amount of clustering in the earlier layers. For AT, collapse undergoes a slower decrease through layers than for ST. Setting: CIFAR-10, $\ell_\infty$ adversary.

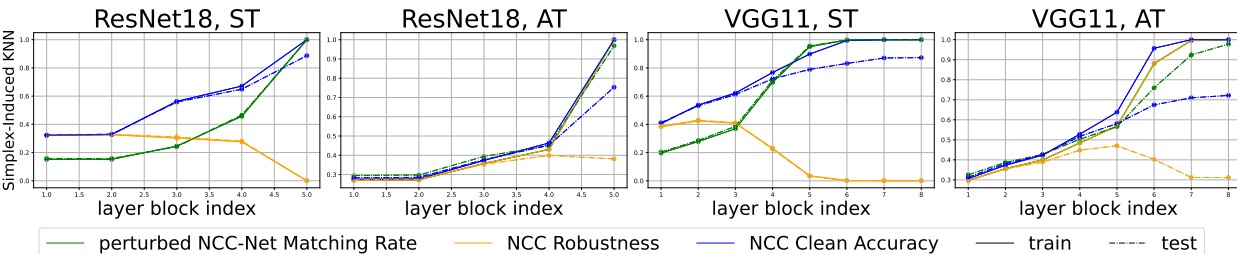

Figure 7: Layerwise NCC classifier. We measure the performance of the NCC classifier obtained from (training) class means on both train and test data. NCC Robustness refers to NCC Accuracy on perturbed data. Note that, on training data, the NCC Robustness and the perturbed NCC-Net Matching Rate curves overlap. Early layers give a surprisingly robust NCC classifier (NCC Robustness) for both train and test data. Setting: CIFAR-10, $\ell_\infty$ adversary.

## 5 Conclusion

Neural Collapse is an interesting phenomenon displayed by classification Neural Networks. We present experiments that quantify the sensitivity of this geometric arrangement to input perturbations, and, further, display that Neural Collapse can appear (but not always does!) in Neural Networks trained to be robust. Specifically, we find Adversarial Training (Madry et al., 2018) induces severe Neural Collapse not only on clean data but also for perturbed data, while TRADES (Zhang et al., 2019) leads to no Neural Collapse at all. Interestingly, simple nearest-neighbors classifiers defined by feature representations (either final or earlier ones) from either standardly or adversarially trained Neural Networks can exhibit remarkable accuracy and

robustness, suggesting robustness is maintained in early layers for both situations, while it diminishes quickly across layers for standardly trained networks. We conclude that Neural Collapse is an elegant phenomenon, prevalent in many deep learning settings, with yet unclear connection to the generalization and robustness of Neural Networks: it is not necessary to appear in adversarially-robust optimization. Our findings on robust networks call for a theoretical analysis of Neural Collapse, possibly moving beyond the unconstrained feature model, which by construction doesn't seem to be able to reason about perturbations in the data.

## Acknowledgments

This work was supported by the National Science Foundation under NSF Award 1922658, the Dean's Undergraduate Research Fund from the NYU College of Arts and Science, and in part through the NYU IT High Performance Computing resources, services, and staff expertise.

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

## Appendix

## A  Exact definitions of NC metrics

**Simplex ETF:** A *standard simplex ETF* composed of $C$ points is a set of points in $\mathbb{R}^C$, each point belonging to a column of

$$\sqrt{\frac{C}{C-1}}(\boldsymbol{I} - \frac{1}{C}\mathbf{1}_C\mathbf{1}_C^\top),$$

where $\boldsymbol{I} \in \mathbb{R}^{C\times C}$ is the identity matrix and $\mathbf{1}_C = \begin{bmatrix} 1 & \cdots & 1 \end{bmatrix}^\top \in \mathbb{R}^C$ is the all-ones vector. In our discussion, a *simplex* can be thought of as a standard simplex ETF up to partial rotations, reflections, and rescaling.

**Between-class and within-class covariance:** Using terminology developed in Section 3, we define between-class covariance $\Sigma_B \in \mathbb{R}^{p\times p}$ as

$$\Sigma_B \triangleq \mathrm{AVG}_c(\boldsymbol{\mu}_c - \boldsymbol{\mu}_G)(\boldsymbol{\mu}_c - \boldsymbol{\mu}_G)^\top$$

and $\Sigma_W \in \mathbb{R}^{p\times p}$ as

$$\Sigma_W \triangleq \mathrm{AVG}_{i,c}(\mathbf{h}_{i,c} - \boldsymbol{\mu}_c)(\mathbf{h}_{i,c} - \boldsymbol{\mu}_c)^\top.$$

**(NC1) Variability Collapse:**

$$\Sigma_B^\dagger\Sigma_W \to \mathbf{0},$$

where $\dagger$ denotes the Moore-Penrose inverse. The NC1 curve corresponds to $\mathrm{Tr}(\Sigma_B^\dagger\Sigma_W)$.

**(NC2) Convergence to Simplex ETF:**

$$\langle \tilde{\boldsymbol{\mu}}_c, \tilde{\boldsymbol{\mu}}_{c'} \rangle \to -\frac{1}{C-1} \quad \forall\ c \neq c'$$

$$|\,\|\boldsymbol{\mu}_c - \boldsymbol{\mu}_G\|_2 - \|\boldsymbol{\mu}_{c'} - \boldsymbol{\mu}_G\|_2\,| \to 0 \quad \forall\ c, c'$$

The NC2 Equinorm curve corresponds to the variation of $\|\boldsymbol{\mu}_c - \boldsymbol{\mu}_G\|_2$ across all labels $c$, the standard deviation of these $c$ quantities: $\mathrm{std}(\|\boldsymbol{\mu}_c - \boldsymbol{\mu}_G\|_2)$. The NC2 Equiangular curve corresponds to $\mathrm{AVG}_{c\neq c'}\ \mathrm{abs}(\langle \tilde{\boldsymbol{\mu}}_c, \tilde{\boldsymbol{\mu}}_{c'} \rangle + \frac{1}{C-1})$, where abs is the absolute value operator.

**(NC3) Convergence to self-duality:**

$$\frac{\mathbf{w}_c}{\|\mathbf{w}_c\|_2} - \frac{\boldsymbol{\mu}_c - \boldsymbol{\mu}_G}{\|\boldsymbol{\mu}_c - \boldsymbol{\mu}_G\|_2} \to 0 \quad \forall\ c.$$

The NC3 curve corresponds to

$$\sqrt{\sum_c \|\frac{\mathbf{w}_c}{\|\mathbf{w}_c\|_2} - \frac{\boldsymbol{\mu}_c - \boldsymbol{\mu}_G}{\|\boldsymbol{\mu}_c - \boldsymbol{\mu}_G\|_2}\|_2^2}.$$

**(NC4) Simplification to NCC classifier:**

$$\arg\max_{c'} \langle \mathbf{w}_{c'}, \mathbf{h} \rangle + b_{c'} \to \arg\min_{c'} \|\mathbf{h} - \boldsymbol{\mu}_{c'}\|_2 \quad \forall\ \mathbf{h} \in H(\mathcal{S}).$$

The NC4 curve corresponds to the mismatch ratio of these two quantities.

**NCC-Network Matching Rate:**

$$\arg\max_{c'} \langle \mathbf{w}_{c'}, \mathbf{h} \rangle + b_{c'} \stackrel{?}{=} \arg\min_{c'} \|\mathbf{h} - \boldsymbol{\mu}_{c'}\|_2, h \in H(\mathcal{S}').$$

This quantity matches NC4 when $S' = S$, where $S$ is the dataset that the classifier was trained on and $S'$ is the dataset of interest. When we study the presence of Neural Collapse in intermediate layers of an already converged network, we first collect its intermediate representations at this layer. The centers of these representations can be used to form a simple NCC classifier. This allows us to compute the alignment of the predictions of the intermediate-layer NCC and the entire network over $S'$.

In our experiments, we calculate the NC statistics with the code provided by Han et al. (2022)[3].

---

[3]https://colab.research.google.com/github/neuralcollapse/neuralcollapse/blob/main/neuralcollapse.ipynb

## B Experimental Details

**Code.** For $\ell_\infty$ and $\ell_2$ PGD attacks with ST and AT, we used the code from Rice et al. (2020a) [4]. For TRADES, we adopted the original implementation[5]. For the 160px ImageNette, all original images have the shortest side resized to 160px. To fulfill the requirement of NC, we use the CenterCrop of 160 to fix the train and test set. We have attached the code for reproducing NC results with ST, AT, and TRADES within a zip file.

**Plotting.** Throughout our paper, we plot all quantities per 5 epochs in all figures.

**Layerwise NC.** We study the layerwise NC1, NC2 and NC4 quantities for both PreActResNet18 (ResNet18) and VGG11. With ResNet18, which consists of one convolutional layer, four residual blocks, and the final linear layer, we use the features after every block for the first five blocks (one convolutional layer and four residual blocks) as representations. With VGG, which consists of eight convolutional blocks (convolutional layer + batch-normalization + max-pooling) and the final linear layer, we use the features after each convolutional block as representations. We apply average-pooling subsampling on representations that are too large for feasible computation of NC1's pseudo-inverse.

## C Complementary Results on CIFAR-10, $\ell_2$ adversary.

Here we complement our main text with robust network experiments on CIFAR-10 for $\ell_2$ adversarial perturbations.

Figure 8 illustrates NC results of Adversarial Training and TRADES training with the $\ell_2$ adversary. All plots are consistent with our findings in the main text: Adversarial Training alters Neural Collapse such that the clean representation simplex overlaps with the perturbed representation simplex, whereas TRADES does not lead to any simplex ETF.

## D Complementary Results on CIFAR-100

In this section, we reproduce our experiments on CIFAR-100. We illustrate results with $(\ell_\infty, \ell_2)$ adversaries and obtain the same conclusions as those on CIFAR-10. This suggests the universality of the intrinsic adversarial perturbation dynamics that we have detailed in the main text.

### D.1 CIFAR-100 $\ell_\infty$ Standard and Adversarial Training Results

All results are summarized within Figure 9. Similar to the main text, we plot the untargeted attack illustration in Figure 10. Notably, on CIFAR-100 with ST, adversarial perturbations also push the representation to leap toward the predicted class's simplex cluster with very small angular deviation.

### D.2 CIFAR-100 $\ell_\infty$ TRADES Results

For CIFAR-100 $\ell_\infty$ trained with TRADES, Figure 11 depicts the results, and we observe that no simplex exists, consistent with previous results.

### D.3 CIFAR-100 $\ell_2$ AT and TRADES Results

These results are shown in Figure 12. All observations are consistent with previous results.

---

[4]https://github.com/locuslab/robust_overfitting
[5]https://github.com/yaodongyu/TRADES

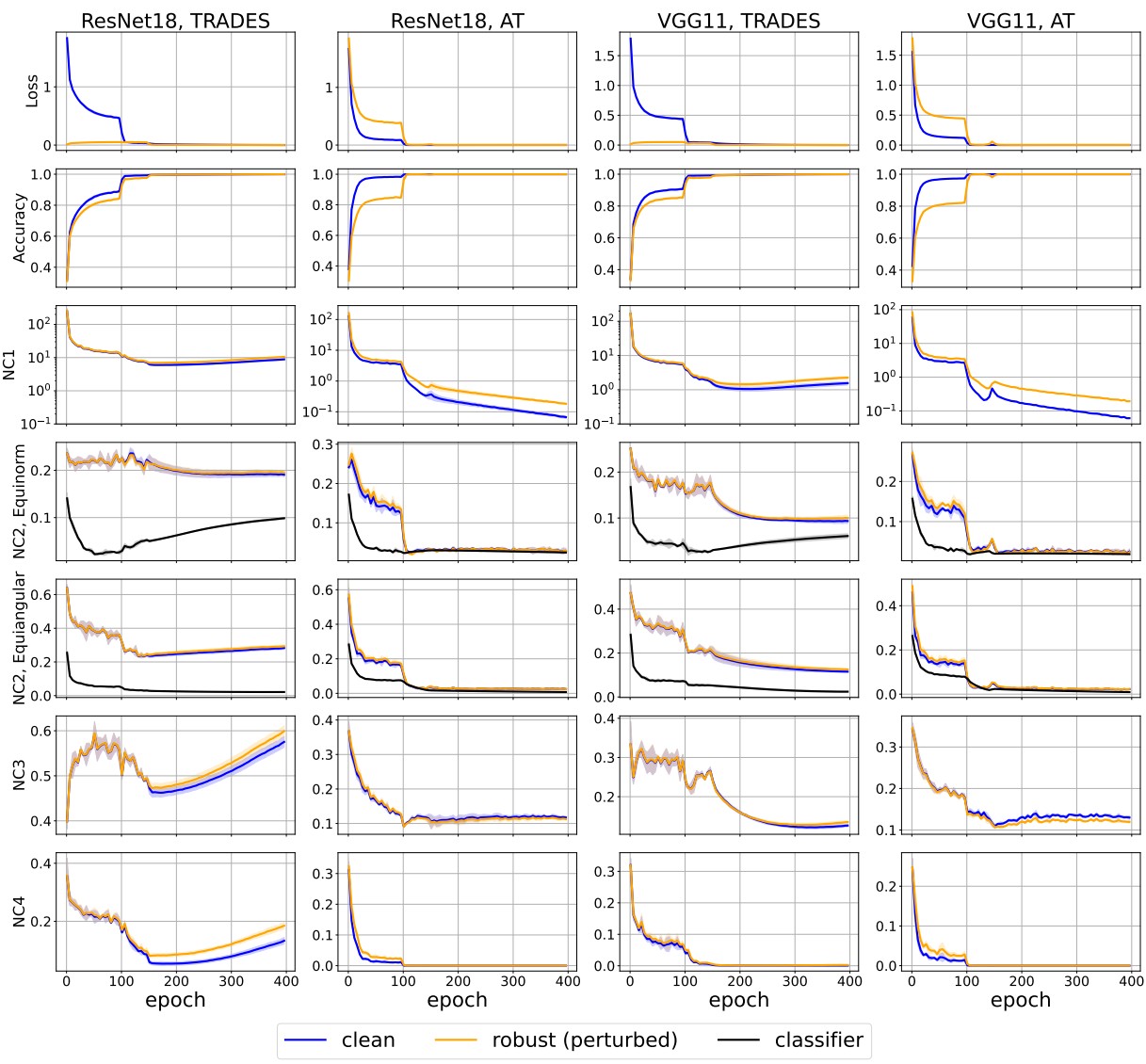

Figure 8: Accuracy, Loss and NC evolution with adversarially trained and TRADES trained networks. Setting: CIFAR-10, $\ell_2$ adversary.

## D.4 CIFAR-100 Simplex Similarity Results

The Simplex Similarity and non-centered Angular Distance of the simplices formed by targeted adversarial and clean examples with ST, and the simplices generated by clean and perturbed examples with AT, are depicted in Figure 13. The result is the same as the one for CIFAR-10 in the main text, Figure 4.

## D.5 CIFAR-100 Layerwise Results

Here in Figure 14 and Figure 15, we perform the same computations as those in Figure 6 and Figure 7. We arrive at the same conclusions.

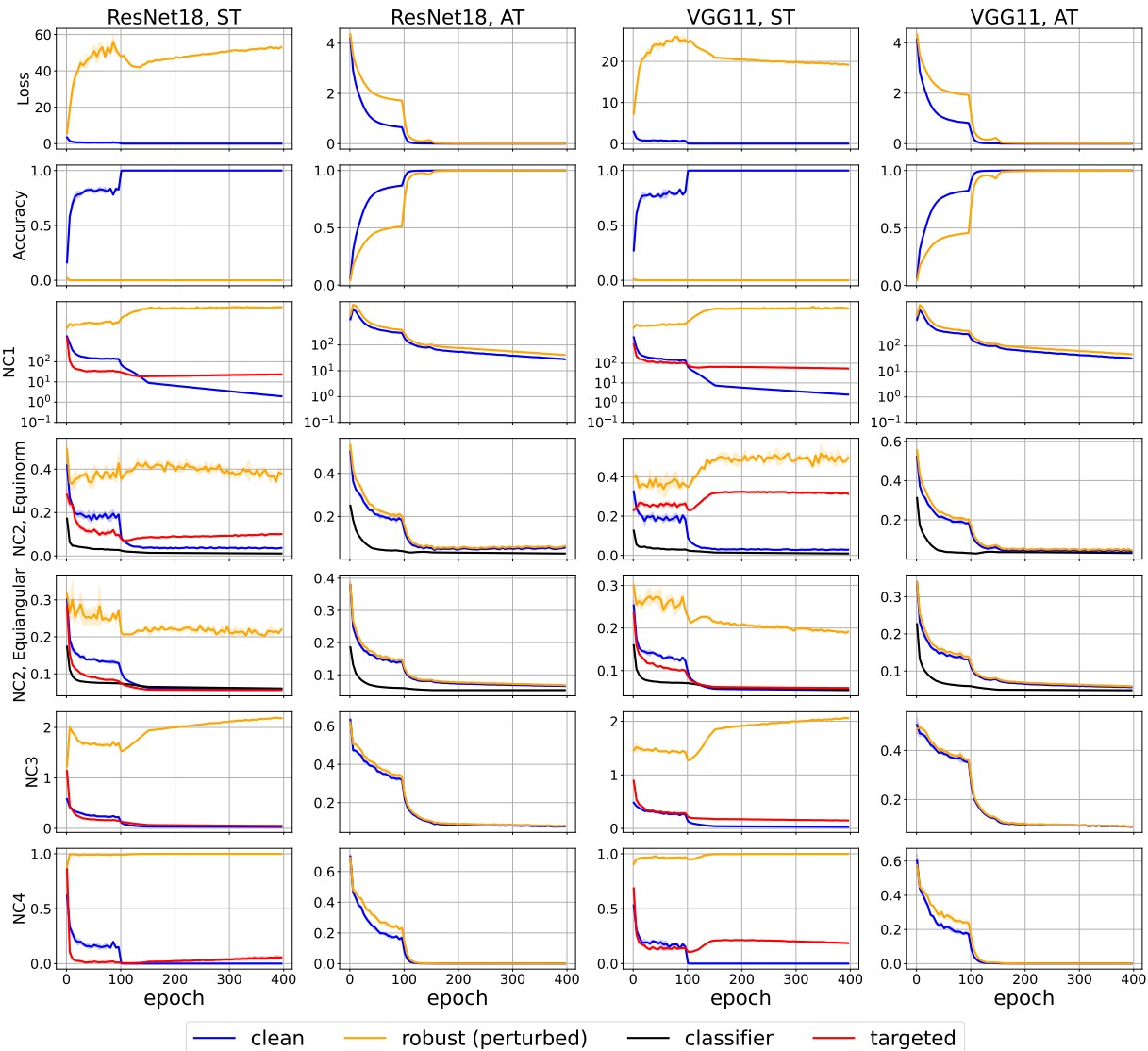

Figure 9: Accuracy, Loss and NC evolution with standardly trained networks. Setting: CIFAR-100, $\ell_\infty$ adversary.

# E  Small Epsilon Results

Here, we illustrate how AT indeed progressively induces more robust NC metrics and simplex ETFs with respect to the perturbation radius $\epsilon$. Figure 16 shows the NC metrics over $8/255-$perturbed data. Conversely, using an ST model, the NC metrics when evaluating on $(2/255, 4/255, 8/255)-$perturbed data also increases monotonically with adversarial strength. This is illustrated in Figure 17.[6]

# F  ImageNette Results

To further corroborate our experimental conclusions, we append ImageNette results here. Results are illustrated in Figure 18, 19, 20 and 21.

---

[6]For small radius AT and small radius adversarial attack for ST, we scale the PGD step size $\alpha$ linearly with $\epsilon$ to ensure PGD to work properly.

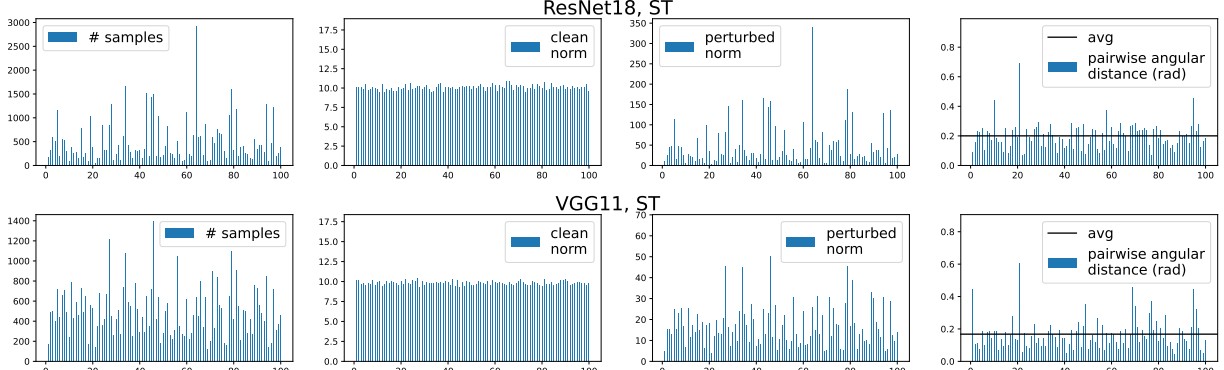

Figure 10: Illustration of untargeted adversarial attacks on standardly trained, converged, models that correspond to one random seed. (CIFAR-100, $\ell_\infty$). *Left:* Number of examples with a certain predicted label. *Inner Left:* The norms of clean class-means. *Inner Right:* The norms of predicted class-means with perturbed data. *Right:* Angular distance between clean and predicted class-mean with perturbed data. *Upper:* ResNet18; *Lower:* VGG11. For 100 classes, the between-class angular distance is $\arccos\left(-\frac{1}{99}\right) = 1.58$ rad $= 90.58$ degrees, while 0.2 rad is only 11.4 degrees.

## G   Test Set Neural Collapse Results

For the completeness of our investigation, we illustrate the test set loss, accuracy, and NC evolution in Figures 22, 23, 24, 25, 26 and 27. We put TRADES and AT results together for a straightforward comparison between these two robust optimization algorithms. Still, TRADES leads to a non-collapsed dynamic, and clearly, representations on perturbed data of robust networks do not form a simplex ETF anymore, because the robust test accuracy is less than 40%. An interesting observation is that though both robust optimization algorithms exhibit significant robust overfitting (Rice et al., 2020b), AT's robust accuracy *increases* in the final stage, with an increment in the NC2 quantity.

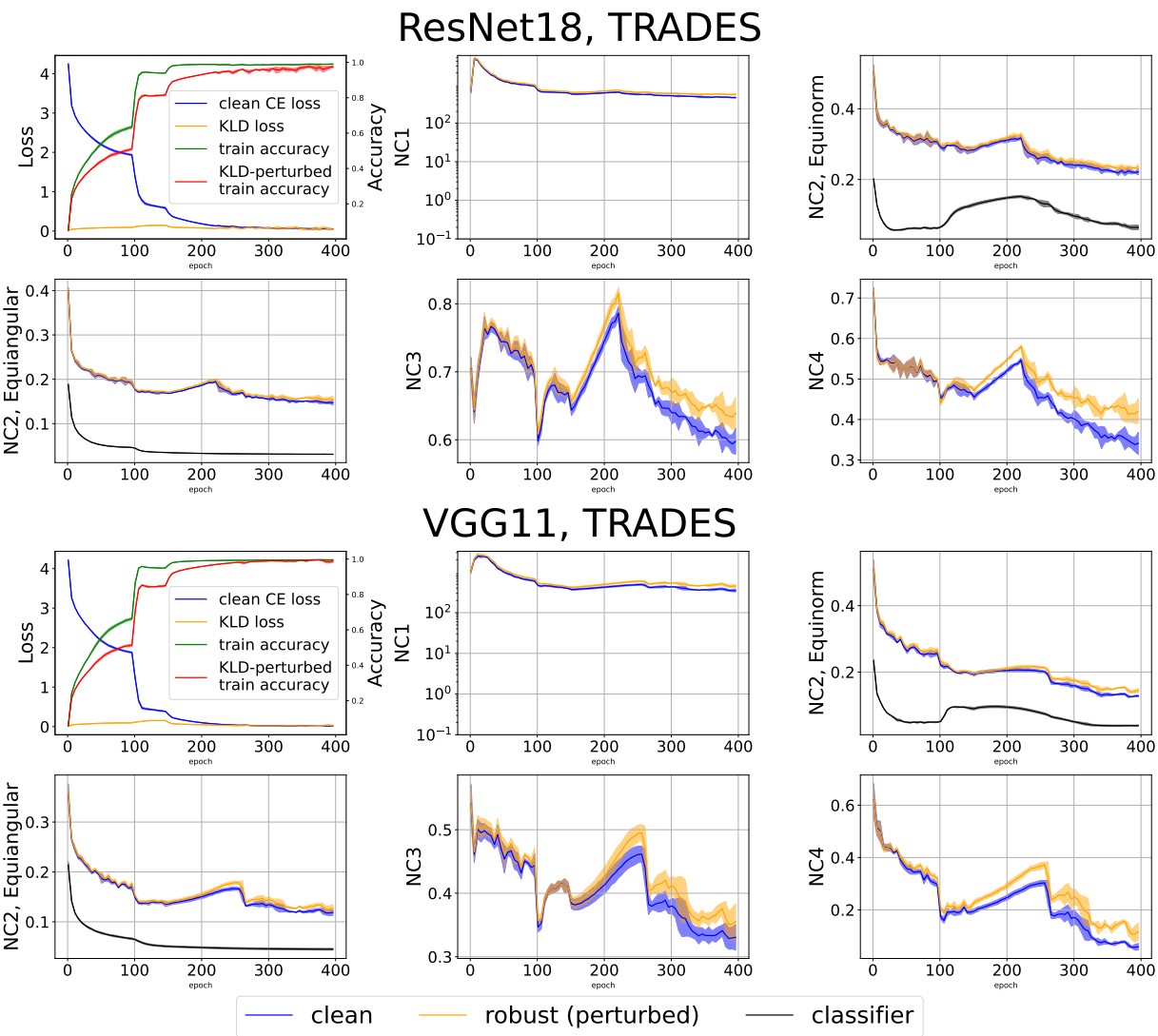

Figure 11: Accuracy, Loss and NC evolution with TRADES trained networks. *Upper:* ResNet18; *Lower:* VGG11. Results indicate AT boosts Neural Collapse so that it also happens on adversarially-perturbed data. Setting: CIFAR-100, $\ell_\infty$ adversary.

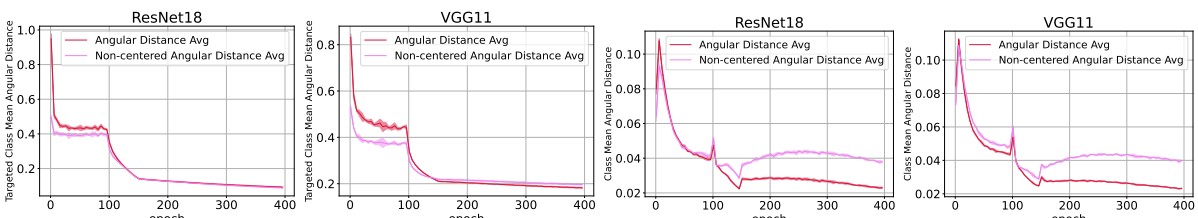

Figure 12: Accuracy, Loss and NC evolution with $\ell_2$ robust models on CIFAR-100. Setting: CIFAR-100, $\ell_2$ adversary.

Figure 13: Angular distance. *Left and Inner Left:* Average between targeted attack class-means and clean class-means on **ST** network. *Inner Right and Right:* Average between perturbed class-means and clean class-means on **AT** network. Setting: CIFAR-100, $\ell_\infty$ adversary.

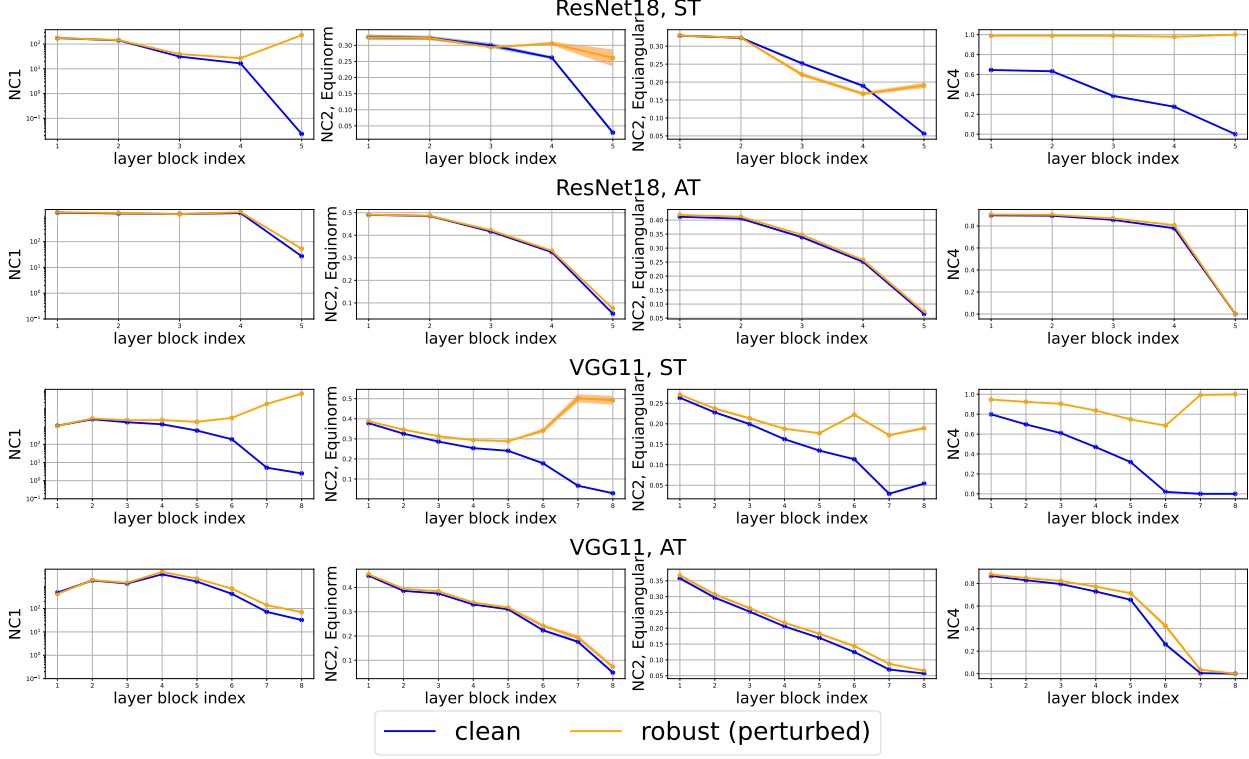

Figure 14: Layerwise evolution of NC1, NC2 and NC4 for ST and AT networks. NC metrics for perturbed data tend to undergo some amount of clustering in the earlier layers. For AT, collapse undergoes a slower decrease through layers than for ST. Setting: CIFAR-100, $\ell_\infty$ adversary.

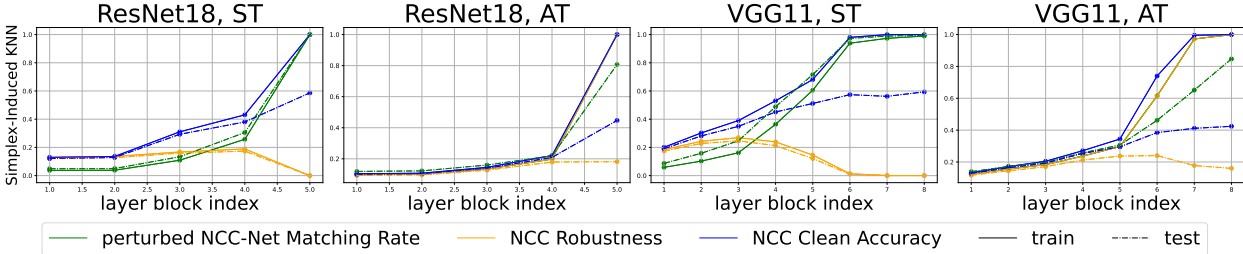

Figure 15: Layerwise NCC classifier. We measure the performance of the NCC classifier obtained from (training) class means on both train and test data. NCC Robustness refers to NCC Accuracy on perturbed data. Note that, on training data, the NCC Robustness and the perturbed NCC-Net Matching Rate curves overlap. Early layers give a surprisingly robust NCC classifier (NCC Robustness) for both train and test data. Setting: CIFAR-100, $\ell_\infty$ adversary.

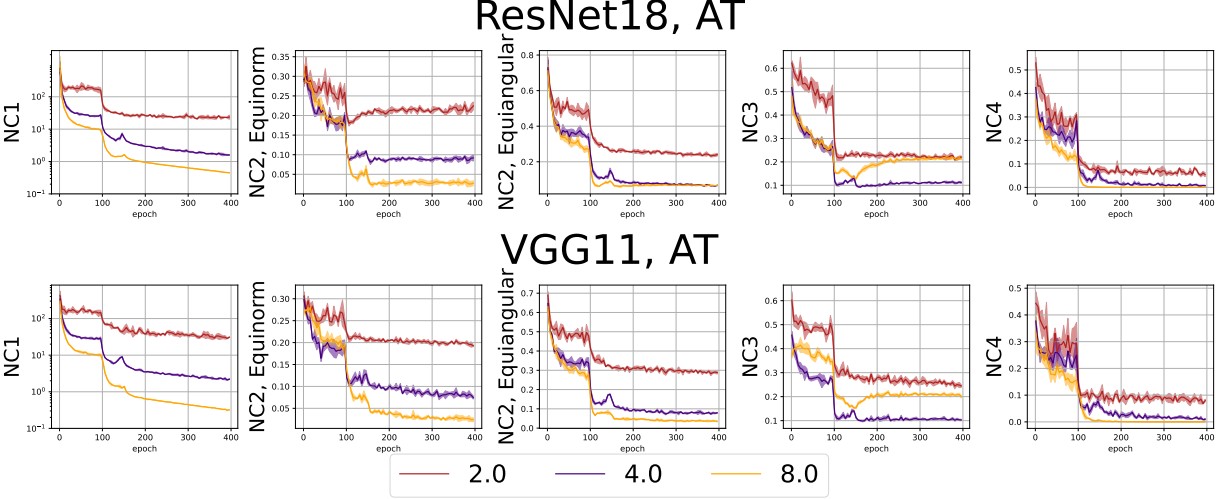

Figure 16: Progressive NC evolution, AT with varying strength. The color indicates the epsilon used for **training**. Setting: CIFAR-10, $\ell_\infty$ adversary.

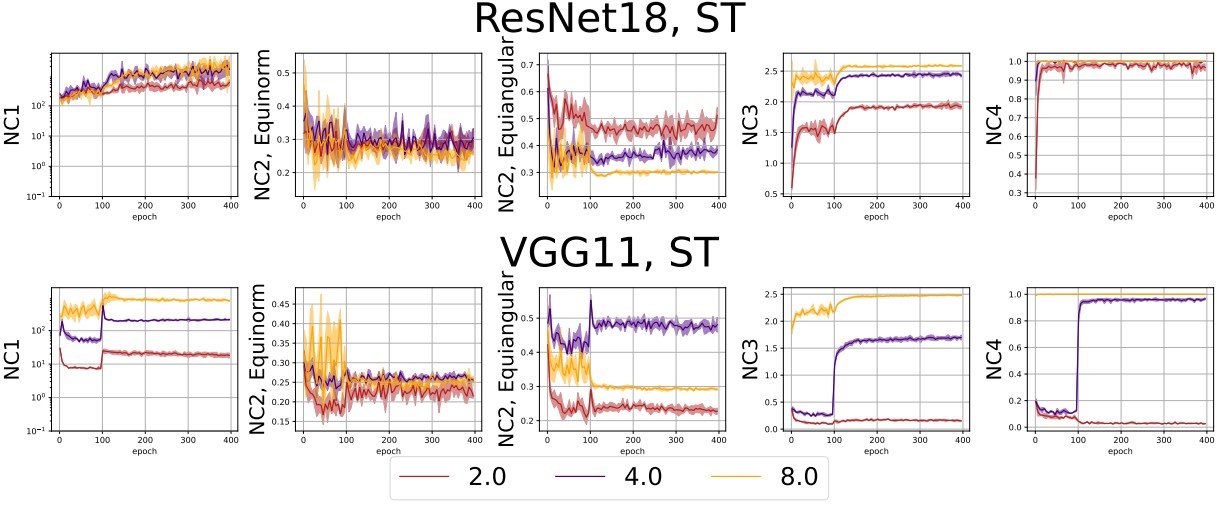

Figure 17: Progressive NC evolution, ST with varying attacking strength. The color indicates the epsilon used for **evaluation**. Setting: CIFAR-10, $\ell_\infty$ adversary.

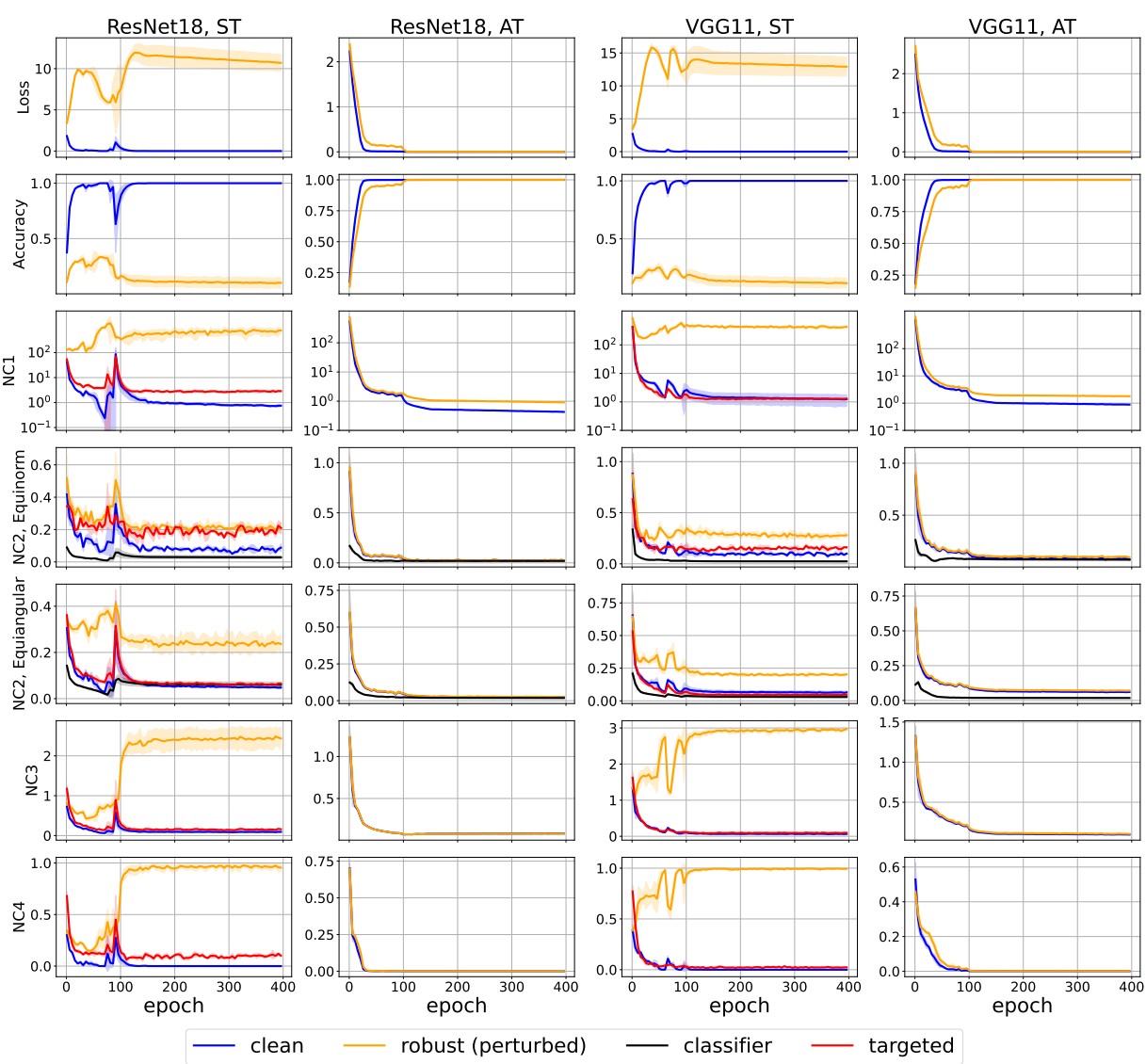

Figure 18: Accuracy, Loss and NC evolution with standardly-trained and adversarially-trained networks. Setting: ImageNette, $\ell_\infty$ adversary.

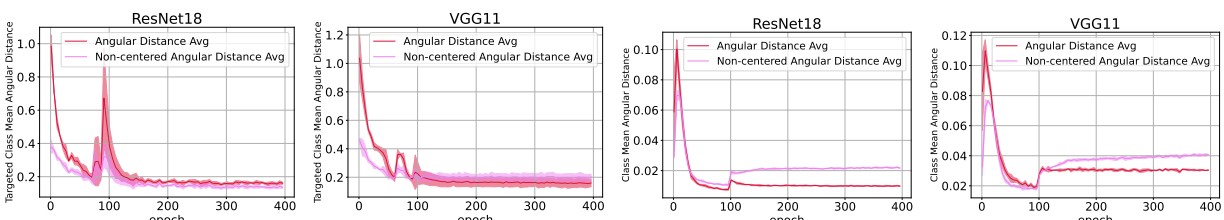

Figure 19: Accuracy, Loss and NC evolution with TRADES trained networks. *Upper:* ResNet18; *Lower:* VGG11. Results indicate AT boosts Neural Collapse so that it also happens on adversarially-perturbed data. Setting: ImageNette, $\ell_\infty$ adversary.

Figure 20: Angular distance. *Left and Inner Left:* Average between targeted attack class-means and clean class-means on **ST** network. *Inner Right and Right:* Average between perturbed class-means and clean class-means on **AT** network. Setting: ImageNette, $\ell_\infty$ adversary.

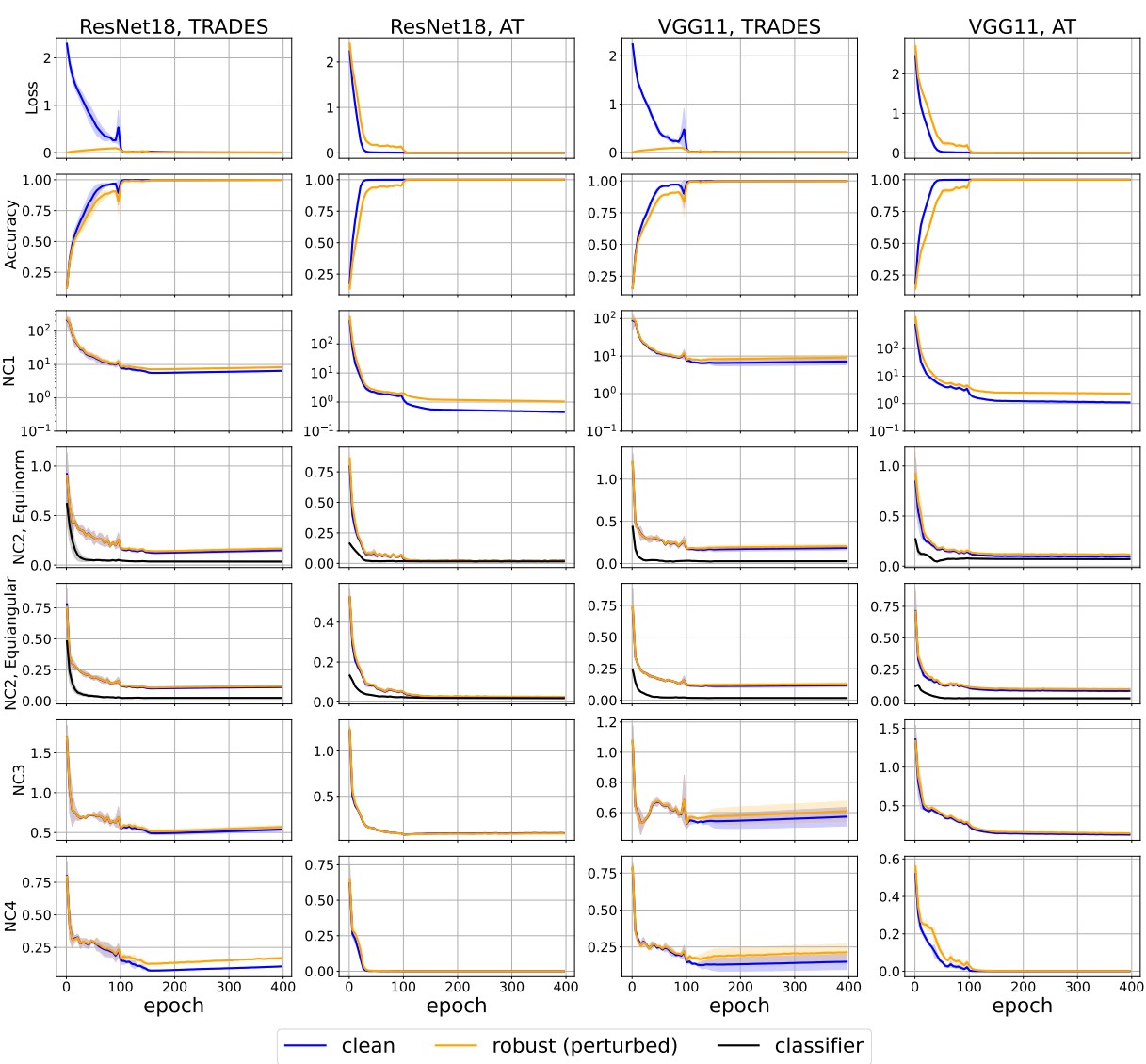

Figure 21: Accuracy, Loss and NC evolution with standardly-trained and adversarially-trained networks. Setting: ImageNette, $\ell_2$ adversary.

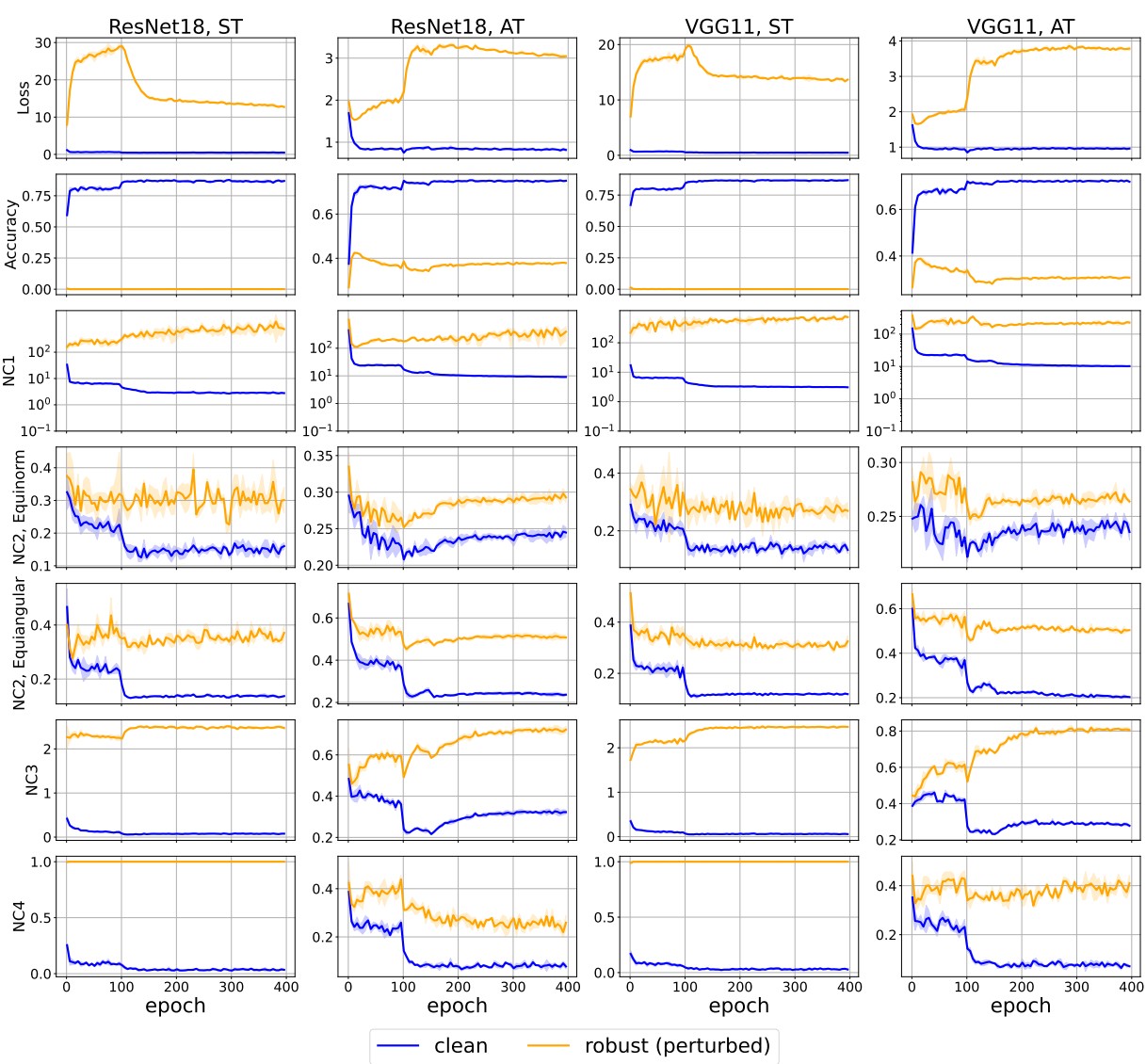

Figure 22: Test set accuracy, Loss and NC evolution with standardly-trained and adversarially-trained networks. Setting: CIFAR-10, $\ell_\infty$ adversary.

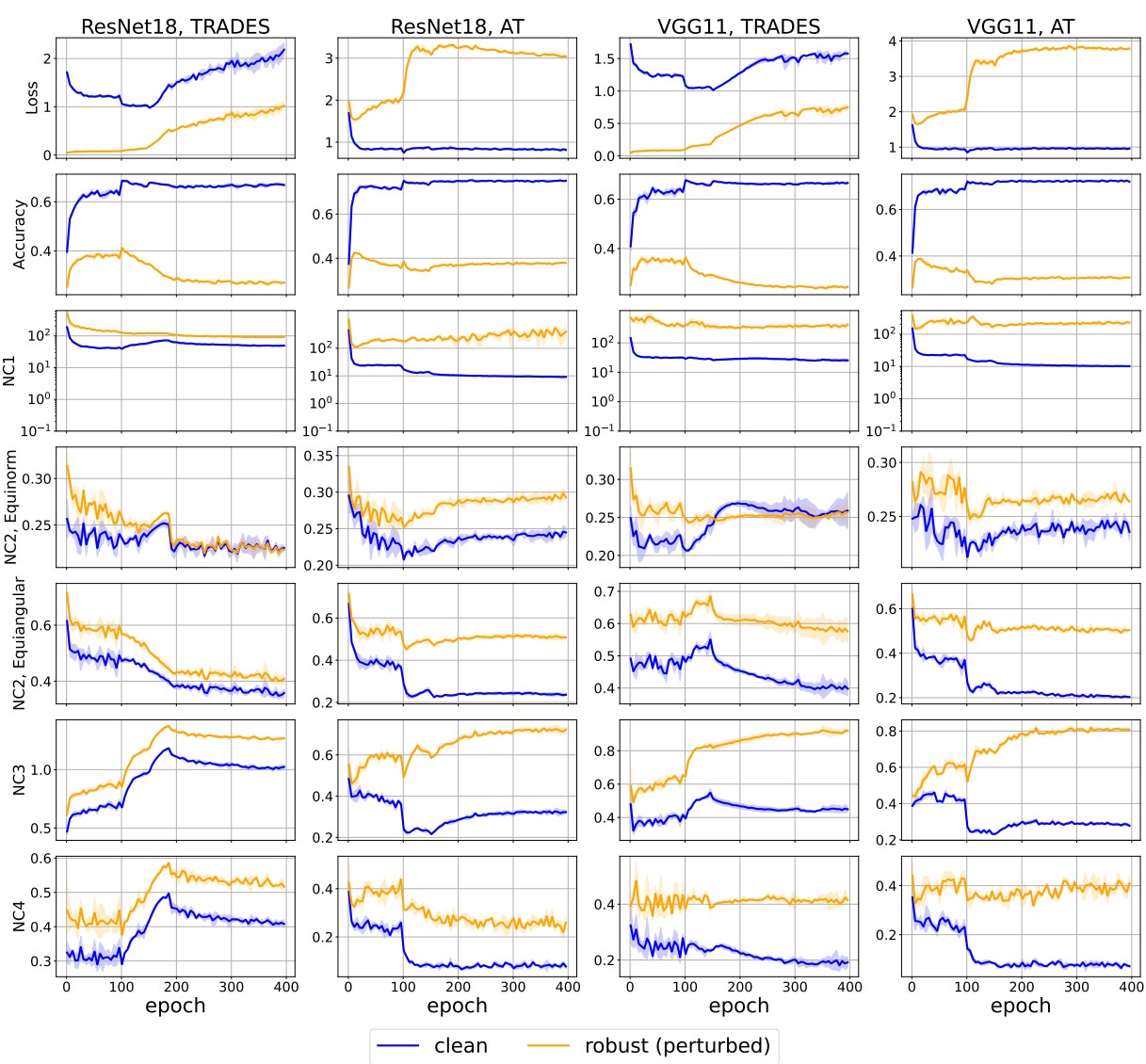

Figure 23: Test set accuracy, Loss and NC evolution with adversarially-trained and TRADES-trained networks. Setting: CIFAR-10, $\ell_\infty$ adversary.

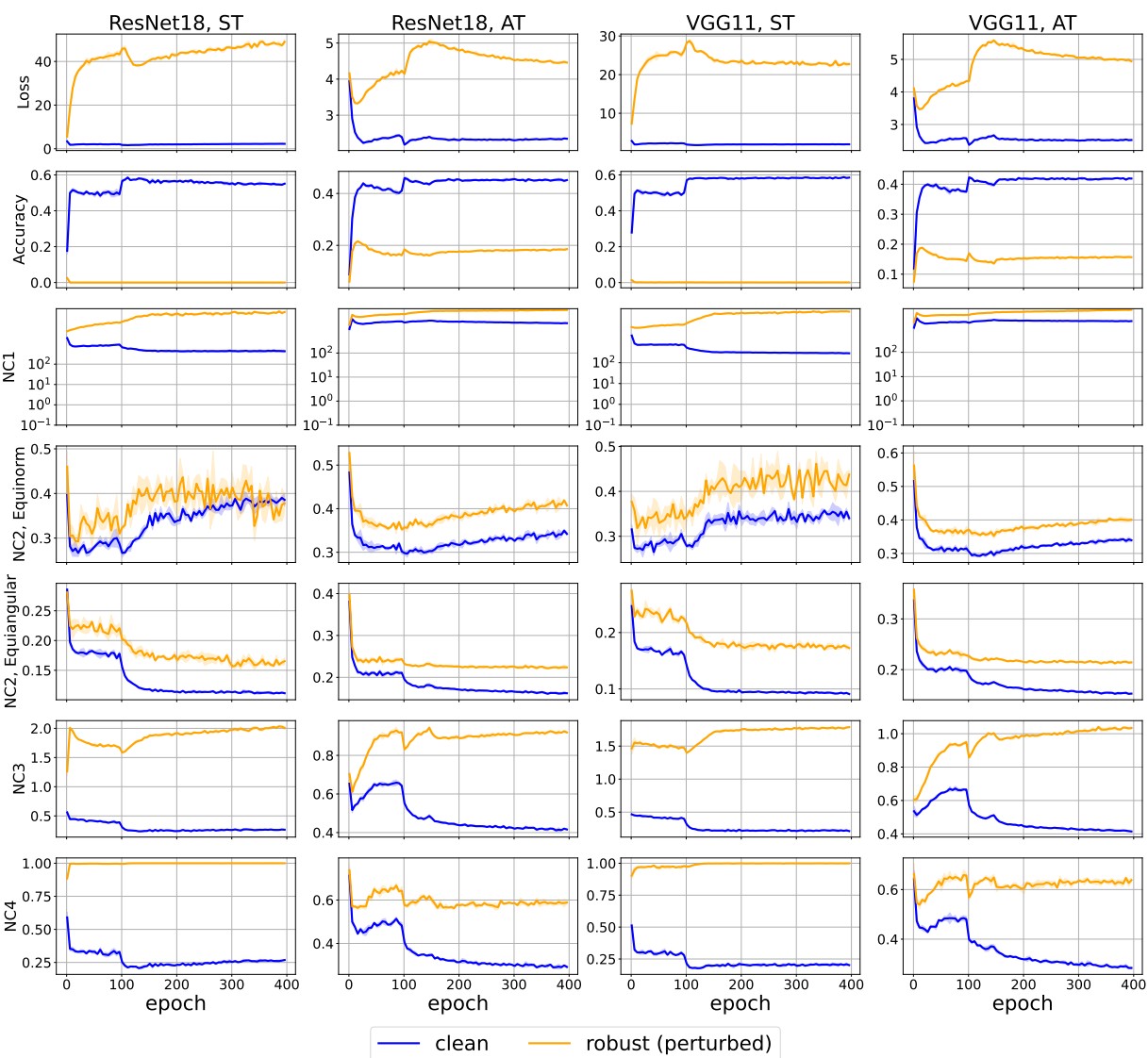

Figure 24: Test set accuracy, Loss and NC evolution with standardly-trained and adversarially-trained networks. Setting: CIFAR-100, $\ell_\infty$ adversary.

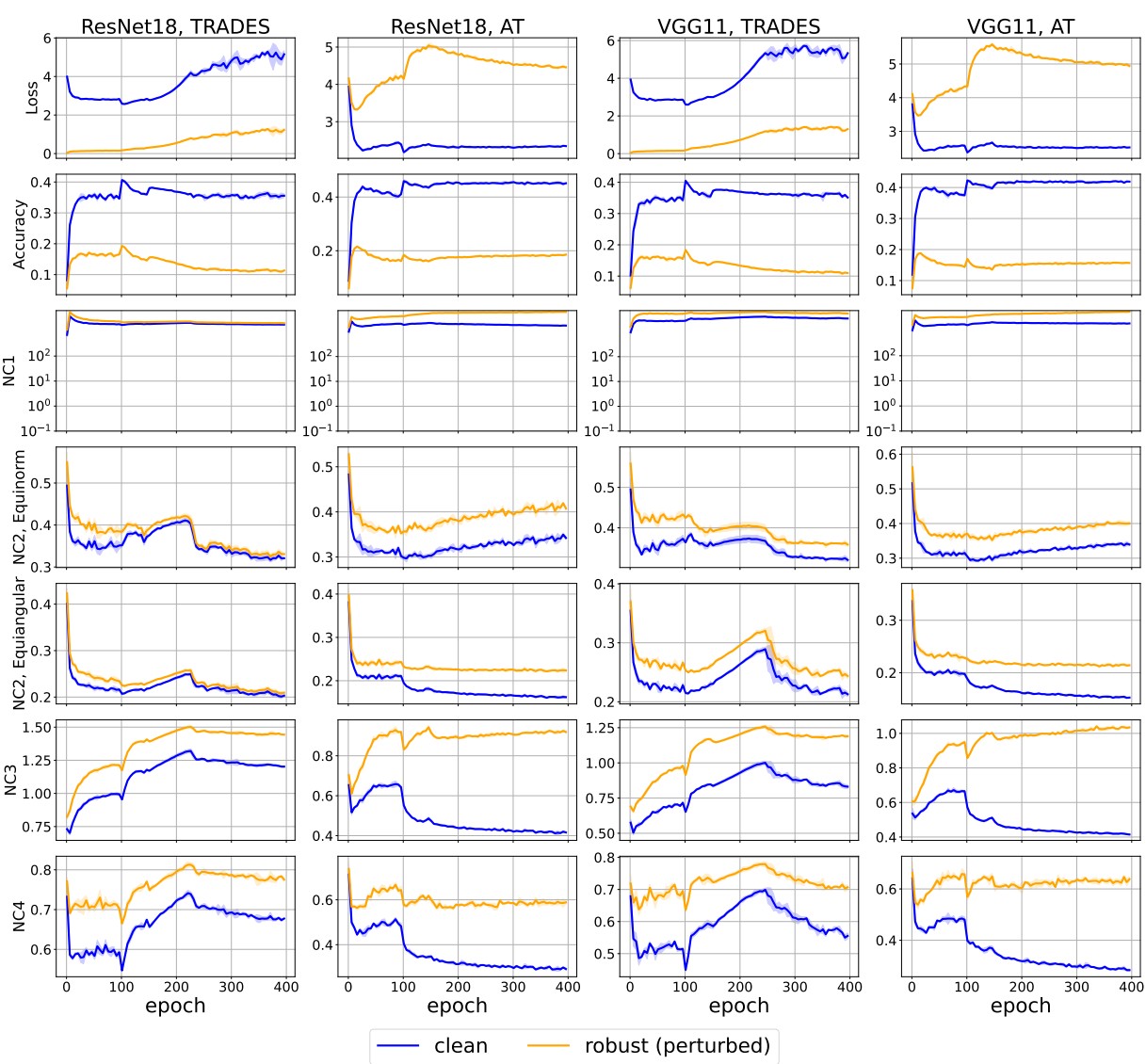

Figure 25: Test set accuracy, Loss and NC evolution with adversarially-trained and TRADES-trained networks. Setting: CIFAR-100, $\ell_\infty$ adversary.

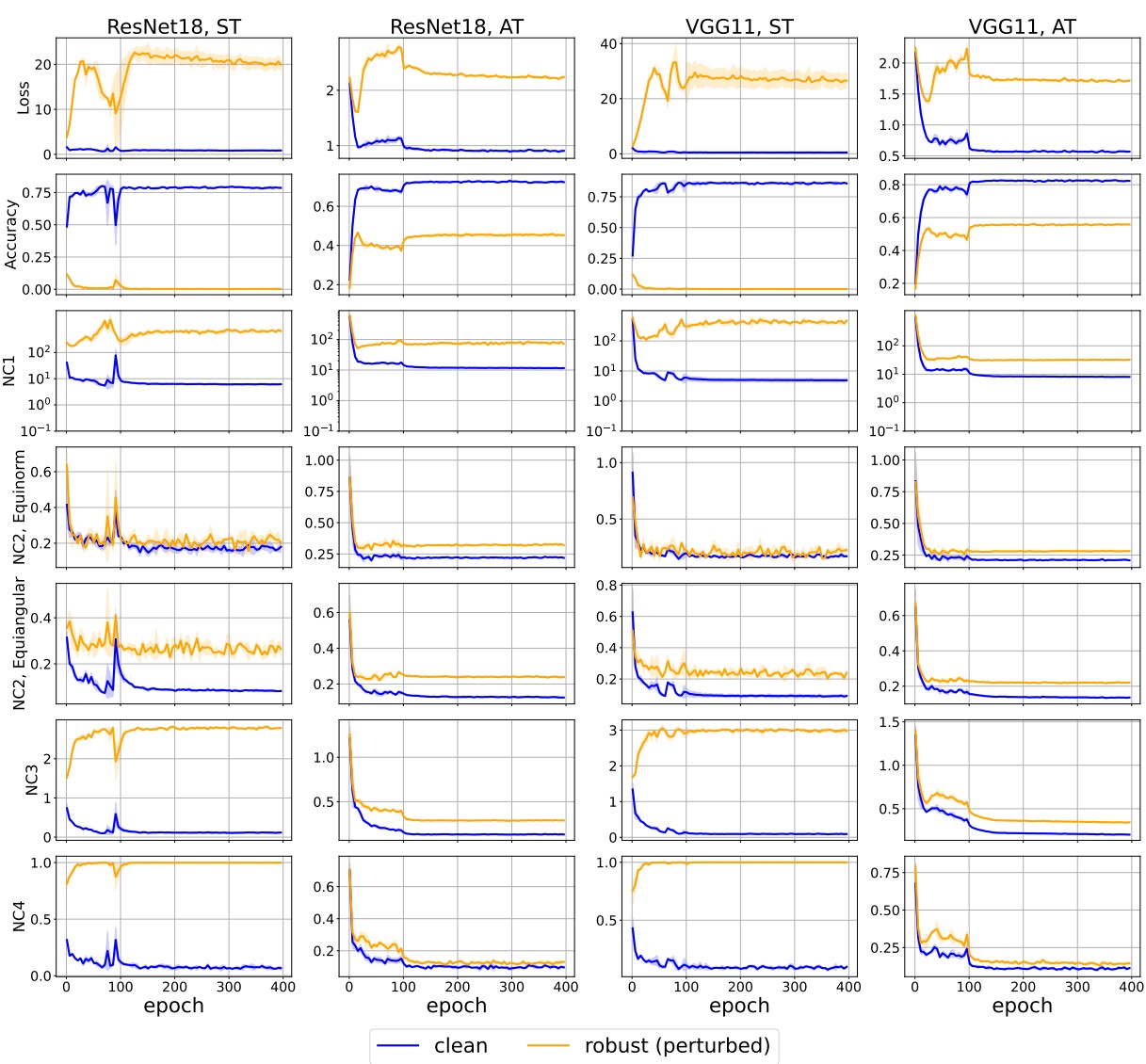

Figure 26: Test set accuracy, Loss and NC evolution with standardly-trained and adversarially-trained networks. Setting: ImageNette, $\ell_\infty$ adversary.

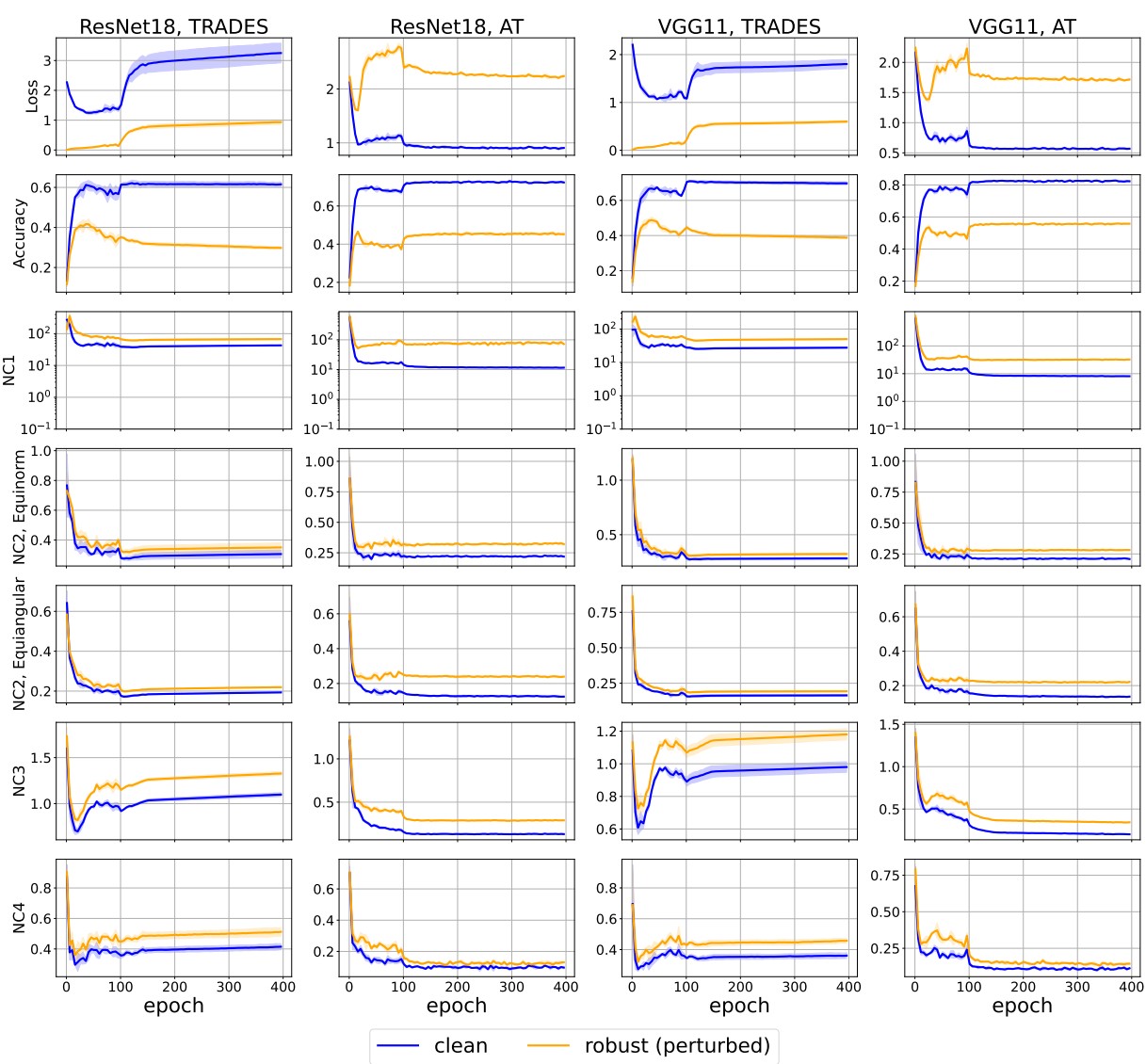

Figure 27: Test set accuracy, Loss and NC evolution with adversarially-trained and TRADES-trained networks. Setting: ImageNette, $\ell_\infty$ adversary.

