# OpenReview forum: "On the Robustness of Neural Collapse and the Neural Collapse of Robustness"
_TMLR — Accepted by TMLR_

### Review · Reviewer_FTau · 2023-12-13

**Summary Of Contributions:**

The authors investigate the relationship between neural collapse and adversarial robustness. Through experiments on standard vision datasets, they find that neural collapse is not robust on standard networks, but is robust for adversarially trained networks. Furthermore, the authors find that earlier layers of the network have more collapse on adversarially perturbed data.

**Audience:**

Yes

**Claims And Evidence:**

Yes

**Requested Changes:**

**Would strengthen work**
* Some of the numbers in the Figures are small and hard to read; it would be great to have them enlarged
* It is not clear what the margins in some of the figures indicate (e.g. Figure 2); it would be helpful to note this in the caption
* Experiments on a larger dataset or different domain

**Strengths And Weaknesses:**

**Strengths**
This paper on the Robustness of Neural Collapse and the Neural Collapse of Robustness has several strengths. Firstly, it investigates a novel connection between neural collapse and robustness, shedding light on the stability properties of simplices and their behavior under adversarial attacks. Secondly, the well-chosen experiments provide comprehensive empirical results that offer valuable insights into the properties of robust and non-robust machine learning models. Notably, the authors consider both TRADES and adversarial training in their experiments and include code for their experiments, which is valuable. Thirdly, the descriptions and illustrations are clear, making the paper accessible to a wide audience. Overall, this paper makes a significant contribution to the field, providing a deeper understanding of the behavior of neural networks and their connection to generalization and robustness.


**Weaknesses**
The main weakness of the paper in my view is that experiments are only performed on CIFAR-10 and CIFAR-100. It would strengthen the paper to also conduct experiments on a larger dataset such as ImageNet or in a different domain (outside vision).

---

> ### Author Response · Authors · 2023-12-28
> **Thanks for your comments!**
>
> Thank you for taking the time to read our paper and your positive review!
>
> Requested changes:
> * *Experiments on a larger dataset or different domain*: Thank you for the suggestion. We selected CIFAR-10 and CIFAR-100 due to their broad usage in the adversarial robustness community, following previous works, including the proposal of adversarial training and TRADES ([1], [2], [3], [4]). Unfortunately, the increased computational cost of adversarial training in comparison to standard training makes ImageNet experiments especially costly in academic environments. In particular, for adversarial training, one needs to first backpropagate through the network K times, where K is the number of PGD steps one uses to generate adversarial examples, and then backpropagate once to update model parameters. This makes adversarial training take a far longer (~K+1) GPU time compared to standard training. Most works that train robust ImageNet models either require fine-tuning pre-trained models [4], which contradicts the problem we are investigating, or propose fast adversarial training variants that only use ~1 PGD step ([5], [6]), which worsens robustness. Alternatively, the work that successfully trained robust ImageNet models directly relies on extremely heavy GPU resources (e.g., in [7], the 8-NVIDIA-A100s with 40GB memory GPU are used to train their robust ImageNet models, leading to 20,000 GPU hours in total). The above facts prevent us from using the complete ImageNet directly for training. However, following your suggestion, we performed our experimental analysis on a 10-class, 160px ImageNet subset, Imagenette [8], that is small enough to train efficiently. We have appended the new results in the revised draft file (Appendix F). The results are consistent with our observations on previous datasets.
> * *Regarding other modalities*: Unfortunately, the field of adversarial robustness mainly considers vision datasets where a notion of adversarial example is well defined (for academic purposes). For language, for example, there are still no widely accepted notions of threat model (like the $\ell_p$ balls for vision). Also, specifically for language, there are no widely accepted notions of a threat model yet. Furthermore, the notion of Neural Collapse is not straightforwardly defined in other modalities. Hence, we decided to stick to what the community considers standard.
>
> * *Some of the numbers in the Figures are small and hard to read* Thank you for the feedback! We enlarged the numbers in the main figures to make them easier to read.
>
> Thank you for helping us improve our work!
>
> [1] Madry et al., Towards deep learning models resistant to adversarial attacks
>
> [2] Zhang et al., Theoretically Principled Trade-off between Robustness and Accuracy
>
> [3] Wu et al., Adversarial Weight Perturbation Helps Robust Generalization
>
> [4] Rice et al., Overfitting in adversarially robust deep learning
>
> [5] Shafahi et al., Adversarial Training for Free!
>
> [6] Wong et al., Fast is better than free: Revisiting adversarial training
>
> [7] Singh et al., Revisiting Adversarial Training for ImageNet: Architectures, Training and Generalization across Threat Models
>
> [8] ImageNette: https://github.com/fastai/imagenette

---

### Review · Reviewer_kJnY · 2023-12-14

**Summary Of Contributions:**

The paper investigates neural collapse phenomena in the context of the adversarially modified examples and adversarially trained networks. It presents empirical investigation based on CIFAR-10 and CIFAR-100 datasets, ResNet18 and VGG11 neural networks and PGD attacks with l_{\inf} and l_2 norms. With the experimental results it is conjectured that not robust networks have a fragile under adversarial attacks simplex of final layer representations, while trained with standard adversarial training approach networks organize into simplex adversarial samples as well. Additionally, it was experimentally analyzed if earlier (shallower) layers exhibit more\less neural collapse behavior on clean and adversarial examples with the conclusion that early layer representations are more robust.

**Audience:**

Yes

**Broader Impact Concerns:**

The work is theoretical and analyses neural collapse for adversarial samples/training. Currently it does not require any impact statement.

**Claims And Evidence:**

No

**Requested Changes:**

The central claim of the paper should be explained better. Is it that adversarial samples form separate simplex and in some cases AT networks also form simplex? Is there anything more? Currently I cannot see a conclusive result that is confirmed throughout the paper and can serve as a contribution. While the experiments are definitely interesting, they mostly contradict each other. The presentation should be adjusted to be more clear and draw visible connection between experimental results and claims. Currently tracking the discussion of experiments back to the original claims is hard. In case this can be clarified and all the claims are indeed confirmed, I would recommend acceptance.

The figures are very hard to read. Maybe it is worth considering leaving less figures in the main text, but making them larger and discussing in more details (additionally referencing to all the other experiments that lead to the same conclusion).

**Strengths And Weaknesses:**

The paper provides large number of experiments and different perspectives on the results obtained. The investigated topic can advance the understanding of adversarial robustness for interpolating networks.

Nevertheless, paper is rather hard to read and it seems that claimed contributions are not supported by experimental results, i.e.,:

I do not understand how the conclusion about samples making "class leap" was made. According to the experiments results description on page 6 and illustration in Figure1 the adversarial samples move a little angle away from the correct class. Why is it called a leap?

The main conjecture about connection between neural collapse and adversarial robustness is confusing: while it is interesting that emergence of neural collapse does not help robustness, it follows from the experiments that robust networks can both have and not have simplex structure (Madry AT VS TRADES). What is the authors conclusion here? Can the answer be traced to some properties of TRADES or standard AT?

It is unclear to me how the conclusions about more neural collapse in the earlier layers for adversarial samples were made: the experiments presented in the paper show that neural collapse is equally absent both in early and late layers for non-robust networks. Correspondingly, the experiments showing robustness of early layer representations seem to be disconnected with the overall line of work. Also, it is claimed in section 4.4 that "earlier layers show same amount of clustering as later on the perturbed data" - but there is no clustering as it is claimed before? And finally, NCC accuracy on clean data matches NCC accuracy on perturbed data according to the experiments - but how high is this accuracy? Is it significant?

Two additional criterion of neural collapse introduced on pages 4-5 are not used in first group of experiments and seems to be needed only for layer-wise analysis (or at least it was not clearly stated). Also, why NCC-Accuracy stems from NC4? NC4 states that classification becomes essentially NCC, but it does not necessarily mean that it is high (low) accuracy.

I do not see a discussion for Figure3 in the text.

There are some formulations in the text that are hard to understand, e.g.:

- "To decouple our analysis from label dependencies that accompany standard attacks" - please explain which dependencies you have in mind. Also please explain why targeted attacks that allow for balanced classes should produce more interesting results.

- What is meant by the "classifier of a class c"? is it a neuron that outputs exactly this class value?

- NC3 means that the weights are becoming equal to representations?

- Definition of NCC-Network matching rate is very strange: in the text it states that it is rate, the formula looks like it is either true or false.

- The criterion of NC should be clearly stated for adversarial samples - currently the text is confusing, using term "perturbed class mean" without defining it

- It is unclear if for experiments with AT networks adversarial samples for generated at each epoch for the state of the network at this epoch or only for the final network? It can severely affect the observed results.

Minor:

- last sentence of caption for figure1 is broken

- S' is not define before using on page 4

- TPT abbreviation is not introduced

- What is the difference between \mu'_c and \mu_{c'}?

---

> ### Author Response · Authors · 2023-12-28
> **Thanks for your comments!**
>
> Thank you for reading our paper and for your comprehensive review. We would address your concerns as follows:
>
> * *I do not understand how the conclusion about samples making "class leap" was made*:
>
>     * The phenomenon of 'cluster leaping' is defined on standardly-trained, non-robust networks. The left-hand side of Figure 1, subtitled 'Standardly-Trained Model,' illustrates this phenomenon. Before applying the adversarial attack in Eq. (1), the representations of all samples concentrate around the correct class cluster. After the attack is performed, the representations 'leap' toward a randomly chosen wrong class cluster. Take the green dots as an example; these training data have nearly identical representations without the attack. However, the representations of the adversarial examples leap toward the blue/red cluster after being attacked because the attack successfully manipulates the network to output class blue/red for them. Thus, the representations of adversarial examples don't move just a little angle but instead leap far away from their original positions.
>     * The results in Figure 4 (page 7), left-hand side, quantitatively measure this behavior. After collecting adversarial examples for each class label, we measure the angular distance between the original class cluster vector and the cluster vector of the corresponding adversarial examples with the same predicted label. The results tell us that this value is very small (~0.1 rad) compared to the between-class angular distance (which is arccos(-1/9) = 1.68 rad), confirming the cluster leaping phenomenon.
>     * We use a targeted attack in Eq. (2) to quantify this behavior because, from Figure 3 and the corresponding illustration of Figure 1 introduced above, we know that without a targeted attack, the number of examples with a certain predicted label could vary a lot or even diminish, since the attack in Eq. (1) does not guarantee the predicted label. This would make the estimation of angular distance biased, as we lack examples for certain classes. Thus, we switch to a targeted attack in Eq. (2), which guarantees a balanced number of examples for each predicted label when the attack success rate approaches 100%.
> * *The main conjecture about connection [...] TRADES or standard AT?*
>     * Neural Collapse has been demonstrated to be prevalent in various settings (such as experiments with cross-entropy [1], MSE loss [2], and in theory [3] [4]). The primary goal of our study in this paper is to explore two key questions: (1) What is the geometry of the representations in non-robust networks after applying an adversarial attack? (2) Does Neural Collapse still happen after adversarial training? We do find empirically that for (2) the answer depends on the loss function, but we do not attempt to provide an explanation in this work since this would be a contribution on its own (see [1, 2]). We believe that our analysis with the NC1-4 metrics is a good starting point for such an investigation.
>
> * *It is unclear to me how the conclusions about more neural collapse in the earlier layers [...] - but how high is this accuracy? Is it significant?*:
>     * The original Neural Collapse (NC) is defined for the last-layer representations. As noted in [5], for non-robust networks, the extent of NC gets larger in deeper layers. This indicates that, for clean data, NC progressively occurs until the very end of the network representation. In Figure 6, the 1st and 3rd rows replicate this finding, showing that the blue line progressively decreases.
>     * Building upon this conclusion, our finding is that for adversarial examples of non-robust networks, although NC does not occur when considering the last layer representation (corresponding to the rightmost point in all subfigures), the extent of NC is quite similar to the clean data's representation when examining the earlier layers. Thus, the conclusion at this stage is not 'NC is equally absent in early layers,' but rather 'surprisingly, the gap of NC extent in the last-layer representation between adversarial examples and clean examples is not observed for earlier layers.' This observation naturally leads us to propose a question: could this similarity in the earlier layers be utilized to extract robustness from non-robust networks? We provide an affirmative answer to this question by constructing the NCC classifier and finding non-trivial robustness in Figure 7. We hope this clarifies how earlier layer robustness is connected to our layerwise NC findings in Figure 6.
>     * Regarding the concern about '*the same amount of “clustering”*': the term 'clustering' we use here refers to the 'extent of NC,' i.e., the NC quantity we plot in Figure 6. There is no true clustering in earlier layers, and that is why we added quotation marks to it. We will rephrase this term to 'the extent of NC' to avoid such confusion.

---

> > ### Author Response · Authors · 2023-12-28
> > **Additional reply to the first comment**
> >
> > * *It is unclear to me how the conclusions about more neural collapse in the earlier layers [...] - but how high is this accuracy? Is it significant?* (cont'd):
> >     * Regarding the significance of accuracy: we do not use data augmentation throughout the paper. For non-robust standardly-trained networks, we can achieve >30% (train and test) robustness with the earlier layer NCC classifier. By comparison, these non-robust networks have 0% (train and test) robustness themselves; and for adversarially-trained networks without augmentation, the best robustness on train/test data is 100%/43%, respectively. Thus, this accuracy is non-trivial and significant.
> > * Two additional criterion [...] Figure3 in the text.
> >     * Yes, these two criteria (NCC-Network Matching Rate and NCC Accuracy) are only used for layerwise analysis. We will clarify this statement within the text.
> >     * Regarding the relationship between NCC Accuracy and NC4: the original proposal of NC focuses on the Terminal Phase of Training, i.e., after the model achieves 100% training accuracy. Within this stage, NCC Accuracy is the same as NC4. It is true that outside of this stage, NCC Accuracy differs from NC4 even when measured on the training set S. This is the reason we refer to this metric as 'stemming from' NC4, as they are not identical.
> >     * As for the discussion for Figure 3, we address it in Section 4.1, specifically within the paragraph titled 'Re-emergence of Simplices: Cluster Leaping.' There, we illustrate the intuition behind the illustrations in Figure 1 and explain why we chose the targeted attack to generate Figure 4, as well as how the cluster leaping phenomenon is derived.
> > * There are some formulations in the text that are hard to understand, e.g.:
> >     * *please explain which dependencies you [...] more interesting results*: To decouple our analysis from label dependencies that accompany standard attacks, we address this dependency in two ways. First, the Neural Collapse measures are proposed by default with ground-truth labels. As demonstrated in the paper, using these default settings hinders our ability to reveal the underlying geometrical structure of representations under adversarial attacks (i.e., the cluster leaping phenomenon). By switching to using the predicted label, we partially recover this structure, as shown in Figure 3. Second, the label-imbalance induced by the standard attack with Eq. (1) introduces estimation bias, as the number of examples with a certain label could vary significantly (see Figure 10 for a more severe illustration). To overcome this issue, we enforce targeted attacks, ensuring a balanced number of examples with respect to predicted labels.
> >     * *What is meant by the "classifier of a class c"? is it a neuron that outputs exactly this class value?*: We define this concept in Section 3.1, where for each class c, the corresponding classifier is denoted by Wc, and the bias by bc. It comes from the final (linear) layer of the network.
> >     * *NC3 means that the weights are becoming equal to representations?*: Yes, NC3 quantifies the extent of alignment between classfication weights and representation cluster means (in a normalized sense).
> >     * *Definition of NCC-Network matching rate is very strange: in the text it states that it is rate, the formula looks like it is either true or false:* We adopt a similar notion as with NC4, where the equation measures whether the prediction of the NCC and the network on a single datum match each other. The rate itself is defined as the matching ratio of these two quantities over a dataset of interest. We will append this description to Appendix A, similar to the description we provide for NC4. Thank you!
> >     * *The criterion of NC should be clearly stated for adversarial samples - currently the text is confusing, using term "perturbed class mean" without defining it*: We use this term for adversarially-robust networks to emphasize that, in the final stage, the examples used to form these means are no longer 'adversarial' since the network could achieve 100% training robustness. We will make this statement clear in the text, with an explicit mention of the dataset concept S we use to construct the simplex, as introduced in Section 3.1. Thank you for the suggestion!
> >     * *It is unclear if for experiments with AT networks adversarial samples for generated at each epoch for the state of the network at this epoch or only for the final network? It can severely affect the observed results.*: As described in Section 4, when collecting feature representations for adversarially perturbed data, we always compute the perturbations at each epoch. This is because NC is defined over the training data, and for adversarially trained models this is the natural option.

---

> > > ### Author Response · Authors · 2023-12-28
> > > **Additional reply to the second comment**
> > >
> > > *  Minor:
> > >     * (1) last sentence of caption for figure1 is broken
> > >     * (2) S' is not define before using on page 4
> > >     * (3) TPT abbreviation is not introduced
> > >     * (4) What is the difference between $\mu^\prime_c$ and $\mu_{c'}$?:
> > >
> > > * Thank you! We fixed the caption and we explicitly state the definition of S’ at the beginning of Section 3. The term 'TPT' is currently mentioned in the footnote; we will move it to the main text to prevent any confusion. Regarding (4), the notation of $\mu_c$ and $\mu^\prime_{c}$ are used to denote the non-normalized class mean vectors associated with datasets S and S’. The $\mu_{c^\prime}$ also refers to a non-normalized class mean vector but is used to avoid confusion with the use of $\mu_c$, where the notation $c^\prime$ always accompanies an argmin/argmax statement. The instances include the $\mu_{c^\prime}$ used in the NCC-Network Matching Rate and the NCC Accuracy.
> > >
> > > [1] Papyan et al., Prevalence of neural collapse during the terminal phase of deep learning training.
> > >
> > > [2] Han et al., Neural Collapse Under MSE Loss: Proximity to and Dynamics on the Central Path.
> > >
> > > [3] Súkeník et al., Deep Neural Collapse Is Provably Optimal for the Deep Unconstrained Features Model.
> > >
> > > [4] Ji et al., An unconstrained layer-peeled perspective on neural collapse.
> > >
> > > [5] He and Su, A law of data separation in deep learning.

---

> > > ### Comment · Reviewer_kJnY · 2023-12-30
> > >
> > > 6 - "please explain which dependencies you [...] more interesting results": So your experiments show that the original labels of adversarial samples would be nearly aligned with original clusters, i.e., small angle away from the mean axis leads to a different prediction? I think it is worth mentioning as a conclusion. Also, please use the term "untargeted attacks" and not "standard attacks".
> > >
> > > 7 - "We define this concept in Section 3.1, where for each class c, the corresponding classifier is denoted by Wc, and the bias by bc. It comes from the final (linear) layer of the network." Please specify what exactly is Wc - if you are given weights W of the last layer, what is Wc?

---

> > > > ### Author Response · Authors · 2024-01-01
> > > > **Reply to additional review and comment**
> > > >
> > > > Thanks for your additional review and comments. We would address your concern as follows:
> > > >
> > > > 1 - There is figure2 on page7, but it does not have angular distances. Do you mean figure4 on page9? If yes, then I do not see there the numbers (0.1 and 1.68) that you are giving as an example.
> > > >
> > > > * Yes, Figure 4 was shifted from page7 to page9 in the revised submission. In particular, the angular distance of approximately 0.1 rad between the targeted attack class-means and clean class-means is depicted in Figure 4, Left and Inner Left. The number of 1.68 rad is derived from NC2-Equiangular (see Sec 3.2), as $\arccos(-\frac{1}{C-1})=\arccos(-\frac{1}{9})=\arccos(\langle\tilde{\mu_c},\tilde{\mu_{c'}}\rangle)=\angle(\tilde{\mu_c}, \tilde{\mu_{c'}}),$ since the vectors are normalized, where $\angle(\cdot,\cdot)$ is the angle between two vectors.
> > > >
> > > > * The angular distance results for untargeted attacks are in Figure 3 and Figure 10 for CIFAR-10 / CIFAR-100, respectively. Here, the average angular distance is ~0.2 rad. We have shown in these figures. that the untargeted attack does not preserve the predicted label balance. Hence, the estimated predicted class-means with fewer examples will have a higher variance.
> > > >
> > > > 2 - Figure1 is just an illustration of your contributions, not a result from any experiment - you cannot make any conclusions from it.
> > > >
> > > > * Yes, we agree that Figure 1 does not lead to direct conclusions. However, Figure 1 is illustrative of the conclusions that we have reached from the observations in Figure 3 and Figure 4. Particular, Figure 1 illustrates the following: (1) the cluster leaping phenomenon; (2) for untargeted attacks, the number of examples in each predicted label varies significantly - many examples could be perturbed towards a certain class while very few are perturbed towards another; (3) for untargeted attacks, the predicted label class-means have varying norms.
> > > >
> > > > 3 - Do you mean that angular distance is biased for untargeted attack because the class means have different estimation precision?
> > > >
> > > > * Yes, and the difference in estimation precision comes from the fact - unlike targeted attacks - we cannot a priori control the number of examples with certain predicted labels. This is seen when comparing Figure 3 to Figure 4: estimation using the examples perturbed by the untargeted attack leads to the angular distance (of the original class cluster and the cluster of adversarial examples with the same predicted label) to be ~0.2 rad, whereas that of the targeted attack is ~0.1 rad.
> > > >
> > > > 4 - So by adversarial training you mean only one particular type of training (from Madry paper)? I then would suggest to specify it in the contributions and moreover maybe specifically discuss that it is unclear if this will hold for any other adversarial training according to your results with TRADES.
> > > >
> > > > * Yes. Adversarial Training (AT) is an optimization algorithm proposed by [1] to produce adversarially robust classifiers. TRADES is another optimization algorithm (with a different objective) that can also produce adversarially robust classifiers. In adversarial robustness literature, AT is the most commonly used and well-cited method to produce robust networks since it directly minimizes the adversarial risk (instead of empirical risk). We will add the (AT) specification.
> > > >
> > > > 5 - Can you please quantitatively describe the extent of NC in earlier layers? How much "weaker" it is than in the deep layers?
> > > >
> > > > * Yes. The quantitative results of layer-wise NC can be seen in Figure 6, following [2]. It is clear that the NC quantities progressively decrease (in a law described by [2]) as the layer gets deeper.
> > > >
> > > > 6 - "please explain which dependencies you [...] more interesting results": So your experiments show that the original labels of adversarial samples would be nearly aligned with original clusters, i.e., small angle away from the mean axis leads to a different prediction? I think it is worth mentioning as a conclusion. Also, please use the term "untargeted attacks" and not "standard attacks".
> > > >
> > > > * We observe that the representation of an adversarial example leaps from the original label's cluster towards the predicted labels' cluster. Take the green dots in Figure 1 as an illustration. After applying the adversarial attack, the representation of an adversarial example with the original label green leaps to the red / blue labels' cluster, and thus be classified as red / blue, respectively. So our experiments show that the representation of adversarial examples would be nearly aligned with the predicted labels' original clusters.
> > > >
> > > > * We will change instances of "standard attack" in our paper to "untargeted attack".

---

> > > > > ### Comment · Reviewer_kJnY · 2024-01-03
> > > > >
> > > > > Thank you for the answers.
> > > > >
> > > > > I only have one more question for the 4 - I rather meant is it even significant in the earlier layers? These numbers that are shown in the plot are hard to connect with intuitive understanding, so I am wondering if there is anything like collapse happening in the early layers, or it is completely absent (obviously, you can still measure all the NC numbers, but their values just will be not significant).

---

> > > > > > ### Author Response · Authors · 2024-01-06
> > > > > > **Reply to review and comment**
> > > > > >
> > > > > > Thanks for your additional comments.
> > > > > >
> > > > > > Regarding your question, it is truly possible that Neural Collapse doesn't occur in earlier layers. Even so, however, NC1 still reflects the "concentration" of the representations quantitatively. [1] reveals the exponential decay law of NC1 across layers, suggesting that deep neural networks learn to gradually separate different labels in the representation space. This is what we mean by "weaker" for earlier layers, as the intra-class variance (i.e. NC1) there is exponentially larger compared to the latter layers. We apply this layerwise analysis to adversarial examples and found that early layer representations have similar NC1 as that of clean examples, even if they are eventually misclassified. This inspired us to attach NCC classifiers on intermediate layers in Figure 7, leading us to find surprisingly high robustness.

---

> > > > > > > ### Comment · Reviewer_kJnY · 2024-01-08
> > > > > > >
> > > > > > > Thank you for clarifications!

---

> > > > ### Author Response · Authors · 2024-01-01
> > > > **Reply to additional review and comment**
> > > >
> > > > 7 - "We define this concept in Section 3.1, where for each class c, the corresponding classifier is denoted by Wc, and the bias by bc. It comes from the final (linear) layer of the network." Please specify what exactly is Wc - if you are given weights W of the last layer, what is Wc?
> > > > * Given weights $W,b$ for the last layer, $w_c$ is the $c$-th row of $W$ and $b_c$ is the $c$-th entry of $b$. We will clarify this in Sec 3.1.
> > > >
> > > > [1] Madry et al., Towards deep learning models resistant to adversarial attacks.
> > > >
> > > > [2] He and Su, A law of data separation in deep learning.

---

> ### Comment · Reviewer_kJnY · 2023-12-30
>
> Thank you for the answers.
>
> 1 - There is figure2 on page7, but it does not have angular distances. Do you mean figure4 on page9? If yes, then I do not see there the numbers (0.1 and 1.68) that you are giving as an example.
>
> 2 - Figure1 is just an illustration of your contributions, not a result from any experiment - you cannot make any conclusions from it.
>
> 3 - Do you mean that angular distance is biased for untargeted attack because the class means have different estimation precision?
>
> 4 - So by adversarial training you mean only one particular type of training (from Madry paper)? I then would suggest to specify it in the contributions and moreover maybe specifically discuss that it is unclear if this will hold for any other adversarial training according to your results with TRADES.
>
> 5 - Can you please quantitatively describe the extent of NC in earlier layers? How much "weaker" it is than in the deep layers?

---

### Review · Reviewer_iSZ4 · 2023-12-18

**Summary Of Contributions:**

This paper empirically studied several properties regarding Neural Collapse on the robustness of NNs and NNs with adversarial training. The paper found that adversarial attacks tend to break Neural Collapse; Neural Collapse also happens in adversarial training but not necessarily in all robust training methods (e.g., TRADES); earlier layers tend to have stronger Neural Collapse.

**Audience:**

Yes

**Claims And Evidence:**

Yes

**Requested Changes:**

See above. Possibly make Section 3 a background section rather than a methodology section.

**Strengths And Weaknesses:**

Strengths:
- This paper analyzed Neural Collapse on NNs under adversarial attacks and NNs with adversarial training. The paper has presented several new and interesting findings along this line.
- Experimental findings sound reasonable and the experiments are detailed.

Weaknesses:
- Section 3 (methodology) seems to be mostly from previous works. If that's the case, I'd suggest the authors make this section as a background section while focusing on the experiments as the major contributions.

---

> ### Author Response · Authors · 2023-12-28
> **Thanks for your comments!**
>
> Regarding requested changes: this section contains both background knowledge (prior NC definitions) and the definitions of measures that we propose in this work to accommodate the analysis of the robustness of NC. We renamed the section to “Background & Methodology”.

---

> > ### Comment · Reviewer_iSZ4 · 2023-12-30
> > **Distinguish background and methodology**
> >
> > Thanks for the revision.
> >
> > But I think it may be better to more clearly distinguish the background and your method in two separate sections, rather than having a "Background & Methodology" section.

---

> > > ### Author Response · Authors · 2024-01-01
> > > **Reply to additional review and comment**
> > >
> > > Thanks for your additional review and comments.
> > >
> > > We follow your advice to rename Section 3 in our previous response because it consists of the NC backgrounds, AT backgrounds, and several new quantities we propose to quantify the proximity of two simplices as well as to study Neural Collapse in intermediate layers. These new quantities would serve as the "methodology" part since the rest are mostly from previous works. We do not segregate the methodology into a standalone section yet due to its succinct nature. However, we could add a sub-subsection in Section 3.2 that contains the methodology only.

---

### Author Response · Authors · 2023-12-28
**General response**

We sincerely thank all reviewers for taking the time to review our paper. We have updated our submission with the red color indicating the revised content. We address the following concerns:

(1) Following Reviewer kJnY and FTau's suggestions, we enlarged the main images' font size to make them easier to read. We also remove the Gaussian benchmark curves to make the Figure clear.

(2) We conduct experiments on ImageNette, a 10-class subset of ImageNet. The results can be found in Appendix F. All conclusions match our previous results on CIFAR-10 and CIFAR-100.

(3) Following the suggestion from Reviewer iSZ4, we renamed section 3 to Background & Methodology.

(4) We explicitly clarify several technical terms, following the advice of Reviewer KJnY.

Thank all reviewers for helping us improve our work.

---

> ### Author Response · Authors · 2024-01-02
> **General response**
>
> We sincerely thank the reviewers again for further comments and suggestions. We have updated our submission once more, highlighting the revised content in red:
>
> (1) We have replaced instances of "standard attack" in the text with "untargeted attack".
>
> (2) In section 3.1, we clarify the relationship between $W$ and $w_c$.
>
> Again, thank you for helping us improve our work.

---

### Decision · Action_Editor_6jqz · 2024-01-27

**Recommendation:** Accept with minor revision

**Comment:**

The authors might want to consider improving the presentation of their conclusions, as multiple reviewers found it difficult to connect some of the experimental results to the conclusions. While the authors made progress towards improving the presentation by explicitly defining technical terms and improving the font-size and layout of figures, I think the authors can clarify their results further in the presentation and writing.

**Audience:**

Researchers interested in the recently discovered phenomenon of neural collapse, as well as researchers who are generally interested in generalization and adversarial robustness might find this work interesting.

**Claims And Evidence:**

This paper studies the phenomenon named neural collapse, and how it interacts with generalization and robustness. Experiments focus on adversarial robustness and adversarial training. The reviewers found that the claims are justified by the experiments. Experiments are performed on CIFAR-10, CIFAR-100, and ImageNette datasets, using ResNet-18 and VGG-11 models.

---

> ### Author Response · Authors · 2024-02-23
> **Camera Ready submission Uploaded**
>
> Dear action editors and reviewers,
>
> we summarize our revision for camera-ready as follows:
>
> (1) we further clarify the discovery of ‘’cluster leaping’' in both the illustrative figure’s caption and within the main context.
>
> (2) we revise our conclusion section to clearly state our technical contribution.
>
> (3) we update the test statistics figure in the appendix to enlarge the font, as well as make the LR decay schedule consistent.
>
> (4) we provide a GitHub link for accessing our code.
>
> We sincerely thank all reviewers and the action editor for suggestions on improving our work.